# Nonparametric von Mises Estimators for Entropies, Divergences and Mutual Informations

**Kirthevasan Kandasamy**
Carnegie Mellon University
kandasamy@cs.cmu.edu

**Akshay Krishnamurthy**
Microsoft Research, NY
akshaykr@cs.cmu.edu

**Barnabás Póczos, Larry Wasserman**
Carnegie Mellon University
bapoczos@cs.cmu.edu, larry@stat.cmu.edu

**James M. Robins**
Harvard University
robins@hsph.harvard.edu

## Abstract

We propose and analyse estimators for statistical functionals of one or more distributions under nonparametric assumptions. Our estimators are derived from the von Mises expansion and are based on the theory of influence functions, which appear in the semiparametric statistics literature. We show that estimators based either on data-splitting or a leave-one-out technique enjoy fast rates of convergence and other favorable theoretical properties. We apply this framework to derive estimators for several popular information theoretic quantities, and via empirical evaluation, show the advantage of this approach over existing estimators.

## 1 Introduction

Entropies, divergences, and mutual informations are classical information-theoretic quantities that play fundamental roles in statistics, machine learning, and across the mathematical sciences. In addition to their use as analytical tools, they arise in a variety of applications including hypothesis testing, parameter estimation, feature selection, and optimal experimental design. In many of these applications, it is important to *estimate* these functionals from data so that they can be used in downstream algorithmic or scientific tasks. In this paper, we develop a recipe for estimating statistical functionals of one or more nonparametric distributions based on the notion of influence functions.

Entropy estimators are used in applications ranging from independent components analysis [15], intrinsic dimension estimation [4] and several signal processing applications [9]. Divergence estimators are useful in statistical tasks such as two-sample testing. Recently they have also gained popularity as they are used to measure (dis)-similarity between objects that are modeled as distributions, in what is known as the "machine learning on distributions" framework [5, 28]. Mutual information estimators have been used in in learning tree-structured Markov random fields [19], feature selection [25], clustering [18] and neuron classification [31]. In the parametric setting, conditional divergence and conditional mutual information estimators are used for conditional two sample testing or as building blocks for structure learning in graphical models. Nonparametric estimators for these quantities could potentially allow us to generalise several of these algorithms to the nonparametric domain. Our approach gives sample-efficient estimators for all these quantities (and many others), which often outperfom the existing estimators both theoretically and empirically.

Our approach to estimating these functionals is based on post-hoc correction of a preliminary estimator using the Von Mises Expansion [7, 36]. This idea has been used before in the semiparametric statistics literature [3, 30]. However, most studies are restricted to functionals of one distribution and have focused on a "data-split" approach which splits the samples for density estimation and functional estimation. While the data-split (DS) estimator is known to achieve the parametric con-

vergence rate for sufficiently smooth densities [3, 14], in practical settings, as we show in our simulations, splitting the data results in poor empirical performance.

In this paper we introduce the method of influence function based nonparametric estimators to the machine learning community and expand on this technique in several novel and important ways. The main contributions of this paper are:

1. We propose a "leave-one-out" (LOO) technique to estimate functionals of a single distribution. We prove that it has the same convergence rates as the DS estimator. However, the LOO estimator has better empirical performance in our simulations since it makes efficient use of the data.

2. We extend both DS and LOO methods to functionals of multiple distributions and analyse their convergence. Under sufficient smoothness both estimators achieve the parametric rate and the DS estimator has a limiting normal distribution.

3. We prove a lower bound for estimating functionals of multiple distributions. We use this to establish minimax optimality of the DS and LOO estimators under sufficient smoothness.

4. We use the approach to construct and implement estimators for various entropy, divergence, mutual information quantities and their conditional versions. A subset of these functionals are listed in Table 1 in the Appendix. Our software is publicly available at `github.com/kirthevasank/if-estimators`.

5. We compare our estimators against several other approaches in simulation. Despite the generality of our approach, our estimators are competitive with and in many cases superior to existing specialised approaches for specific functionals. We also demonstrate how our estimators can be used in machine learning applications via an image clustering task.

Our focus on information theoretic quantities is due to their relevance in machine learning applications, rather than a limitation of our approach. Indeed our techniques apply to any smooth functional.

**History:** We provide a brief history of the post-hoc correction technique and influence functions. We defer a detailed discussion of other approaches to estimating functionals to Section 5. To our knowledge, the first paper using a post-hoc correction estimator was that of Bickel and Ritov [2]. The line of work following this paper analysed integral functionals of a single one dimensional density of the form $\int \nu(p)$ [2, 3, 11, 14]. A recent paper by Krishnamurthy et al. [12] also extends this line to functionals of multiple densities, but only considers polynomial functionals of the form $\int p^\alpha q^\beta$ for densities $p$ and $q$. All approaches above of use data splitting. Our work contributes to this line of research in two ways: we extend the technique to a more general class of functionals and study the empirically superior LOO estimator.

A fundamental quantity in the design of our estimators is the *influence function*, which appears both in robust and semiparametric statistics. Indeed, our work is inspired by that of Robins et al. [30] and Emery et al. [6] who propose a (data-split) influence-function based estimator for functionals of a single distribution. Their analysis for nonparametric problems rely on ideas from semiparametric statistics: they define influence functions for parametric models and then analyse estimators by looking at all parametric submodels through the true parameter.

## 2 Preliminaries

Let $\mathcal{X}$ be a compact metric space equipped with a measure $\mu$, e.g. the Lebesgue measure. Let $F$ and $G$ be measures over $\mathcal{X}$ that are absolutely continuous w.r.t $\mu$. Let $f, g \in L_2(\mathcal{X})$ be the Radon-Nikodym derivatives with respect to $\mu$. We focus on estimating functionals of the form:

$$T(F) = T(f) = \phi\left(\int \nu(f)d\mu\right) \qquad \text{or} \qquad T(F, G) = T(f, g) = \phi\left(\int \nu(f, g)d\mu\right), \quad (1)$$

where $\phi, \nu$ are real valued Lipschitz functions that twice differentiable. Our framework permits more general functionals (e.g. functionals based on the conditional densities), but we will focus on this form for ease of exposition. To facilitate presentation of the main definitions, it is easiest to work with functionals of one distribution $T(F)$. Define $\mathcal{M}$ to be the set of all measures that are absolutely continuous w.r.t $\mu$, whose Radon-Nikodym derivatives belong to $L_2(\mathcal{X})$.

Central to our development is the Von Mises expansion (VME), which is the distributional analog of the Taylor expansion. For this we introduce the Gâteaux derivative which imposes a notion of differentiability in topological spaces. We then introduce the *influence function*.

**Definition 1.** *Let $P, H \in \mathcal{M}$ and $U : \mathcal{M} \to \mathbb{R}$ be any functional. The map $U' : \mathcal{M} \to \mathbb{R}$ where $U'(H; P) = \frac{\partial U(P+tH)}{\partial t}\big|_{t=0}$ is called the **Gâteaux derivative** at $P$ if the derivative exists and is linear and continuous in $H$. $U$ is Gâteaux differentiable at $P$ if the Gâteaux derivative exists at $P$.*

**Definition 2.** *Let $U$ be Gâteaux differentiable at $P$. A function $\psi(\cdot; P) : \mathcal{X} \to \mathbb{R}$ which satisfies $U'(Q - P; P) = \int \psi(x; P) \mathrm{d}Q(x)$, is the **influence function** of $U$ w.r.t the distribution $P$.*

By the Riesz representation theorem, the influence function exists uniquely since the domain of $U$ is a bijection of $L_2(\mathcal{X})$ and consequently a Hilbert space. The classical work of Fernholz [7] defines the influence function in terms of the Gâteaux derivative by,

$$\psi(x; P) = U'(\delta_x - P; P) = \frac{\partial U((1-t)P + t\delta_x)}{\partial t}\bigg|_{t=0}, \tag{2}$$

where $\delta_x$ is the dirac delta function at $x$. While our functionals are defined only on non-atomic distributions, we can still use (2) to compute the influence function. The function computed this way can be shown to satisfy Definition 2.

Based on the above, the first order VME is,

$$U(Q) = U(P) + U'(Q - P; P) + R_2(P, Q) = U(P) + \int \psi(x; P) \mathrm{d}Q(x) + R_2(P, Q), \tag{3}$$

where $R_2$ is the second order remainder. Gâteaux differentiability alone will not be sufficient for our purposes. In what follows, we will assign $Q \to F$ and $P \to \widehat{F}$, where $F, \widehat{F}$ are the true and estimated distributions. We would like to bound the remainder in terms of a distance between $F$ and $\widehat{F}$. For functionals $T$ of the form (1), we restrict the domain to be only measures with continuous densities, Then, we can control $R_2$ using the $L_2$ metric of the densities. This essentially means that our functionals satisfy a stronger form of differentiability called Fréchet differentiability [7, 36] in the $L_2$ metric. Consequently, we can write all derivatives in terms of the densities, and the VME reduces to a functional Taylor expansion on the densities (Lemmas 9, 10 in Appendix A):

$$T(q) = T(p) + \phi'\left(\int \nu(p)\right) \int (q - p)\nu'(p) + R_2(p, q)$$
$$= T(p) + \int \psi(x; p)q(x)\mathrm{d}\mu(x) + \mathcal{O}(\|p - q\|_2^2). \tag{4}$$

This expansion will be the basis for our estimators.

These ideas generalise to functionals of multiple distributions and to settings where the functional involves quantities other than the density. A functional $T(P, Q)$ of two distributions has two Gâteaux derivatives, $T_i'(\cdot; P, Q)$ for $i = 1, 2$ formed by perturbing the $i$th argument with the other fixed. The influence functions $\psi_1, \psi_2$ satisfy, $\forall P_1, P_2 \in \mathcal{M}$,

$$T_1'(Q_1 - P_1; P_1, P_2) = \frac{\partial T(P_1 + t(Q_1 - P_1), P_2)}{\partial t}\bigg|_{t=0} = \int \psi_1(u; P_1, P_2)\mathrm{d}Q_1(u), \tag{5}$$

$$T_2'(Q_2 - P_2; P_1, P_2) = \frac{\partial T(P_1, P_2 + t(Q_2 - P_2))}{\partial t}\bigg|_{t=0} = \int \psi_2(u; P_1, P_2)\mathrm{d}Q_2(u).$$

The VME can be written as,

$$T(q_1, q_2) = T(p_1, p_2) + \int \psi_1(x; p_1, p_2)q_1(x)\mathrm{d}x + \int \psi_2(x; p_1, p_2)q_2(x)\mathrm{d}x$$
$$+ \mathcal{O}(\|p_1 - q_1\|_2^2) + \mathcal{O}(\|p_2 - q_2\|_2^2). \tag{6}$$

## 3 Estimating Functionals

First consider estimating a functional of a single distribution, $T(f) = \phi(\int \nu(f)d\mu)$ from samples $X_1^n \sim f$. We wish to find an estimator $\widehat{T}$ with low expected mean squared error (MSE) $\mathbb{E}[(\widehat{T} - T)^2]$.

Using the VME (4), Emery et al. [6] and Robins et al. [30] suggest a natural estimator. If we use half of the data $X_1^{n/2}$ to construct an estimate $\hat{f}^{(1)}$ of the density $f$, then by (4):

$$T(f) - T(\hat{f}^{(1)}) = \int \psi(x; \hat{f}^{(1)}) f(x) d\mu + \mathcal{O}(\|f - \hat{f}^{(1)}\|_2^2).$$

As the influence function does not depend on (the unknown) $F$, the first term on the right hand side is simply an expectation of $\psi(X; \hat{f}^{(1)})$ w.r.t $F$. We can use the second half of the data $X_{n/2+1}^n$ to estimate this expectation with its sample mean. This leads to the following preliminary estimator:

$$\widehat{T}_{\mathrm{DS}}^{(1)} = T(\hat{f}^{(1)}) + \frac{1}{n/2} \sum_{i=n/2+1}^{n} \psi(X_i; \hat{f}^{(1)}). \tag{7}$$

We can similarly construct an estimator $\widehat{T}_{\mathrm{DS}}^{(2)}$ by using $X_{n/2+1}^n$ for density estimation and $X_1^{n/2}$ for averaging. Our final estimator is obtained via $\widehat{T}_{\mathrm{DS}} = (\widehat{T}_{\mathrm{DS}}^{(1)} + \widehat{T}_{\mathrm{DS}}^{(2)})/2$. In what follows, we shall refer to this estimator as the Data-Split (DS) estimator. The DS estimator for functionals of one distribution has appeared before in the statistics literature [2, 3, 30].

The rate of convergence of this estimator is determined by the $\mathcal{O}(\|f - \hat{f}^{(1)}\|_2^2)$ error in the VME and the $n^{-1}$ rate for estimating an expectation. Lower bounds from several literature [3, 14] confirm minimax optimality of the DS estimator when $f$ is sufficiently smooth. The data splitting trick is common approach [3, 12, 14] as the analysis is straightforward. While in theory DS estimators enjoy good rates of convergence, data splitting is unsatisfying from a practical standpoint since using only half the data each for estimation and averaging invariably decreases the accuracy.

To make more effective use of the sample, we propose a Leave-One-Out (LOO) version of the above estimator,

$$\widehat{T}_{\mathrm{LOO}} = \frac{1}{n} \sum_{i=1}^{n} \left( T(\hat{f}_{-i}) + \psi(X_i; \hat{f}_{-i}) \right). \tag{8}$$

where $\hat{f}_{-i}$ is a density estimate using all the samples $X_1^n$ except for $X_i$. We prove that the LOO Estimator achieves the same rate of convergence as the DS estimator but empirically performs much better. Our analysis is specialised to the case where $\hat{f}_{-i}$ is a kernel density estimate (Section 4).

We can extend this method to estimate functionals of two distributions. Say we have $n$ i.i.d samples $X_1^n$ from $f$ and $m$ samples $Y_1^m$ from $g$. Akin to the one distribution case, we propose the following DS and LOO versions.

$$\widehat{T}_{\mathrm{DS}}^{(1)} = T(\hat{f}^{(1)}, \hat{g}^{(1)}) + \frac{1}{n/2} \sum_{i=n/2+1}^{n} \psi_f(X_i; \hat{f}^{(1)}, \hat{g}^{(1)}) + \frac{1}{m/2} \sum_{j=m/2+1}^{m} \psi_g(Y_j; \hat{f}^{(1)}, \hat{g}^{(1)}). \tag{9}$$

$$\widehat{T}_{\mathrm{LOO}} = \frac{1}{\max(n,m)} \sum_{i=1}^{\max(n,m)} \left( T(\hat{f}_{-i}, \hat{g}_{-i}) + \psi_f(X_i; \hat{f}_{-i}, \hat{g}_{-i}) + \psi_g(Y_i; \hat{f}_{-i}, \hat{g}_{-i}) \right). \tag{10}$$

Here, $\hat{g}^{(1)}, \hat{g}_{-i}$ are defined similar to $\hat{f}^{(1)}, \hat{f}_{-i}$. For the DS estimator, we swap the samples to compute $\widehat{T}_{\mathrm{DS}}^{(2)}$ and average. For the LOO estimator, if $n > m$ we cycle through the points $Y_1^m$ until we have summed over all $X_1^n$ or vice versa. $\widehat{T}_{\mathrm{LOO}}$ is asymmetric when $n \neq m$. A seemingly natural alternative would be to sum over all $nm$ pairings of $X_i$'s and $Y_j$'s. However, this is computationally more expensive. Moreover, a straightforward modification of our proof in Appendix D.2 shows that both approaches converge at the same rate if $n$ and $m$ are of the same order.

**Examples:** We demonstrate the generality of our framework by presenting estimators for several entropies, divergences mutual informations and their conditional versions in Table 1 (Appendix H). For many functionals in the table, *these are the first computationally efficient estimators proposed*. We hope this table will serve as a good reference for practitioners. For several functionals (e.g. conditional and unconditional Rényi-$\alpha$ divergence, conditional Tsallis-$\alpha$ mutual information) the estimators are not listed only because the expressions are too long to fit into the table. Our software implements a total of 17 functionals which include all the estimators in the table. In Appendix F we illustrate how to apply our framework to derive an estimator for any functional via an example.

As will be discussed in Section 5, when compared to other alternatives, our technique has several favourable properties: the computational complexity of our method is $O(n^2)$ when compared to $O(n^3)$ of other methods; for several functionals we do not require numeric integration; unlike most other methods [28, 32], we do not require any tuning of hyperparameters.

## 4  Analysis

Some smoothness assumptions on the densities are warranted to make estimation tractable. We use the Hölder class, which is now standard in nonparametrics literature.

**Definition 3.** *Let $\mathcal{X} \subset \mathbb{R}^d$ be a compact space. For any $r = (r_1, \ldots, r_d), r_i \in \mathbb{N}$, define $|r| = \sum_i r_i$ and $D^r = \frac{\partial^{|r|}}{\partial x_1^{r_1} \ldots \partial x_d^{r_d}}$. The **Hölder class** $\Sigma(s, L)$ is the set of functions on $L_2(\mathcal{X})$ satisfying,*

$$|D^r f(x) - D^r f(y)| \leq L\|x - y\|^{s-r},$$

*for all $r$ s.t. $|r| \leq \lfloor s \rfloor$ and for all $x, y \in \mathcal{X}$.*

Moreover, define the Bounded Hölder Class $\Sigma(s, L, B', B)$ to be $\{f \in \Sigma(s, L) : B' < f < B\}$. Note that large $s$ implies higher smoothness. Given $n$ samples $X_1^n$ from a $d$-dimensional density $f$, the kernel density estimator (KDE) with bandwidth $h$ is $\hat{f}(t) = 1/(nh^d) \sum_{i=1}^n K\left(\frac{t-X_i}{h}\right)$. Here $K : \mathbb{R}^d \to \mathbb{R}$ is a smoothing kernel [35]. When $f \in \Sigma(s, L)$, by selecting $h \in \Theta(n^{\frac{-1}{2s+d}})$ the KDE achieves the minimax rate of $\mathcal{O}_P(n^{\frac{-2s}{2s+d}})$ in mean squared error. Further, if $f$ is in the bounded Hölder class $\Sigma(s, L, B', B)$ one can truncate the KDE from below at $B'$ and from above at $B$ and achieve the same convergence rate [3]. In our analysis, the density estimators $\hat{f}^{(1)}, \hat{f}_{-i}, \hat{g}^{(1)}, \hat{g}_{-i}$ are formed by either a KDE or a truncated KDE, and we will make use of these results.

We will also need the following regularity condition on the influence function. This is satisfied for smooth functionals including those in Table 1. We demonstrate this in our example in Appendix F.

**Assumption 4.** *For a functional $T(f)$ of one distribution, the influence function $\psi$ satisfies,*

$$\mathbb{E}\left[(\psi(X; f') - \psi(X; f))^2\right] \in \mathcal{O}(\|f - f'\|^2) \quad as \quad \|f - f'\|^2 \to 0.$$

*For a functional $T(f, g)$ of two distributions, the influence functions $\psi_f, \psi_g$ satisfy,*

$$\mathbb{E}_f\left[(\psi_f(X; f', g') - \psi_f(X; f, g))^2\right] \in \mathcal{O}(\|f - f'\|^2 + \|g - g'\|^2) \quad as \quad \|f - f'\|^2, \|g - g'\|^2 \to 0.$$

$$\mathbb{E}_g\left[(\psi_g(Y; f', g') - \psi_g(Y; f, g))^2\right] \in \mathcal{O}(\|f - f'\|^2 + \|g - g'\|^2) \quad as \quad \|f - f'\|^2, \|g - g'\|^2 \to 0.$$

Under the above assumptions, Emery et al. [6], Robins et al. [30] show that the DS estimator on a single distribution achieves MSE $\mathbb{E}[(\widehat{T}_{\mathrm{DS}} - T(f))^2] \in \mathcal{O}(n^{\frac{-4s}{2s+d}} + n^{-1})$ and further is asymptotically normal when $s > d/2$. Their analysis in the semiparametric setting contains the nonparametric setting as a special case. In Appendix B we review these results with a simpler self contained analysis that directly uses the VME and has more interpretable assumptions. An attractive property of our proof is that it is agnostic to the density estimator used provided it achieves the correct rates.

For the LOO estimator (Equation (8)), we establish the following result.

**Theorem 5** (**Convergence of LOO Estimator for** $T(f)$). *Let $f \in \Sigma(s, L, B, B')$ and $\psi$ satisfy Assumption 4. Then, $\mathbb{E}[(\widehat{T}_{\mathrm{LOO}} - T(f))^2]$ is $\mathcal{O}(n^{\frac{-4s}{2s+d}})$ when $s < d/2$ and $\mathcal{O}(n^{-1})$ when $s \geq d/2$.*

The key technical challenge in analysing the LOO estimator (when compared to the DS estimator) is in bounding the variance as there are several correlated terms in the summation. The bounded difference inequality is a popular trick used in such settings, but this requires a supremum on the influence functions which leads to significantly worse rates. Instead we use the Efron-Stein inequality which provides an integrated version of bounded differences that can recover the correct rate when coupled with Assumption 4. Our proof is contingent on the use of the KDE as the density estimator. While our empirical studies indicate that $\widehat{T}_{\mathrm{LOO}}$'s limiting distribution is normal (Fig 2(c)), the proof seems challenging due to the correlation between terms in the summation. We conjecture that $\widehat{T}_{\mathrm{LOO}}$ is indeed asymptotically normal but for now leave it to future work.

We reiterate that while the convergence rates are the same for both DS and LOO estimators, the data splitting degrades empirical performance of $\widehat{T}_{\mathrm{DS}}$ as we show in our simulations.

Now we turn our attention to functionals of two distributions. When analysing asymptotics we will assume that as $n, m \to \infty$, $n/(n+m) \to \zeta \in (0, 1)$. Denote $N = n + m$. For the DS estimator (9) we generalise our analysis for one distribution to establish the theorem below.

**Theorem 6** (**Convergence/Asymptotic Normality of DS Estimator for** $T(f, g)$). *Let* $f, g \in \Sigma(s, L, B, B')$ *and* $\psi_f, \psi_g$ *satisfy Assumption 4. Then,* $\mathbb{E}[(\widehat{T}_{\mathrm{DS}} - T(f, g))^2]$ *is* $\mathcal{O}(n^{\frac{-4s}{2s+d}} + m^{\frac{-4s}{2s+d}})$ *when* $s < d/2$ *and* $\mathcal{O}(n^{-1} + m^{-1})$ *when* $s \geq d/2$. *Further, when* $s > d/2$ *and when* $\psi_f, \psi_g \neq \mathbf{0}$, $\widehat{T}_{\mathrm{DS}}$ *is asymptotically normal,*

$$\sqrt{N}(\widehat{T}_{\mathrm{DS}} - T(f, g)) \xrightarrow{\mathcal{D}} \mathcal{N}\left(0, \frac{1}{\zeta}\mathbb{V}_f[\psi_f(X; f, g)] + \frac{1}{1 - \zeta}\mathbb{V}_g[\psi_g(Y; f, g)]\right). \quad (11)$$

The convergence rate is analogous to the one distribution case with the estimator achieving the parametric rate under similar smoothness conditions. The asymptotic normality result allows us to construct asymptotic confidence intervals for the functional. Even though the asymptotic variance of the influence function is not known, by Slutzky's theorem any consistent estimate of the variance gives a valid asymptotic confidence interval. In fact, we can use an influence function based estimator for the asymptotic variance, since it is also a differentiable functional of the densities. We demonstrate this in our example in Appendix F.

The condition $\psi_f, \psi_g \neq \mathbf{0}$ is somewhat technical. When *both* $\psi_f$ and $\psi_g$ are zero, the first order terms vanishes and the estimator converges very fast (at rate $1/n^2$). However, the asymptotic behavior of the estimator is unclear. While this degeneracy occurs only on a meagre set, it does arise for important choices, such as the null hypothesis $f = g$ in two-sample testing problems.

Finally, for the LOO estimator (10) on two distributions we have the following result. Convergence is analogous to the one distribution setting and the parametric rate is achieved when $s > d/2$.

**Theorem 7** (**Convergence of LOO Estimator for** $T(f, g)$). *Let* $f, g \in \Sigma(s, L, B, B')$ *and* $\psi_f, \psi_g$ *satisfy Assumption 4. Then,* $\mathbb{E}[(\widehat{T}_{\mathrm{LOO}} - T(f, g))^2]$ *is* $\mathcal{O}(n^{\frac{-4s}{2s+d}} + m^{\frac{-4s}{2s+d}})$ *when* $s < d/2$ *and* $\mathcal{O}(n^{-1} + m^{-1})$ *when* $s \geq d/2$.

For many functionals, a Hölderian assumption ($\Sigma(s, L)$) alone is sufficient to guarantee the rates in Theorems 5,6 and 7. However, for some functionals (such as the $\alpha$-divergences) we require $\hat{f}, \hat{g}, f, g$ to be bounded above and below. Existing results [3, 12] demonstrate that estimating such quantities is difficult without this assumption.

Now we turn our attention to the question of statistical difficulty. Via lower bounds given by Birgé and Massart [3] and Laurent [14] we know that the DS and LOO estimators are minimax optimal when $s > d/2$ for functionals of one distribution. In the following theorem, we present a lower bound for estimating functionals of two distributions.

**Theorem 8** (**Lower Bound for** $T(f, g)$). *Let* $f, g \in \Sigma(s, L)$ *and* $\widehat{T}$ *be any estimator for* $T(f, g)$. *Define* $\tau = \min\{8s/(4s + d), 1\}$. *Then there exists a strictly positive constant* $c$ *such that,*

$$\liminf_{n \to \infty} \inf_{\widehat{T}} \sup_{f, g \in \Sigma(s, L)} \mathbb{E}[(\widehat{T} - T(f, g))^2] \geq c\left(n^{-\tau} + m^{-\tau}\right).$$

Our proof, given in Appendix E, is based on LeCam's method [35] and generalises the analysis of Birgé and Massart [3] for functionals of one distribution. This establishes minimax optimality of the DS/LOO estimators for functionals of two distributions when $s \geq d/2$. However, when $s < d/2$ there is a gap between our upper and lower bounds. It is natural to ask if it is possible to improve on our rates in this regime. A series of work [3, 11, 14] shows that, for integral functionals of one distribution, one can achieve the $n^{-1}$ rate when $s > d/4$ by estimating the second order term in the functional Taylor expansion. This second order correction was also done for polynomial functionals of two distributions with similar statistical gains [12]. While we believe this is possible here, these estimators are conceptually complicated and computationally expensive – requiring $O(n^3 + m^3)$ running time compared to the $O(n^2 + m^2)$ running time for our estimator. The first order estimator has a favorable balance between statistical and computational efficiency. Further, not much is known about the limiting distribution of second order estimators.

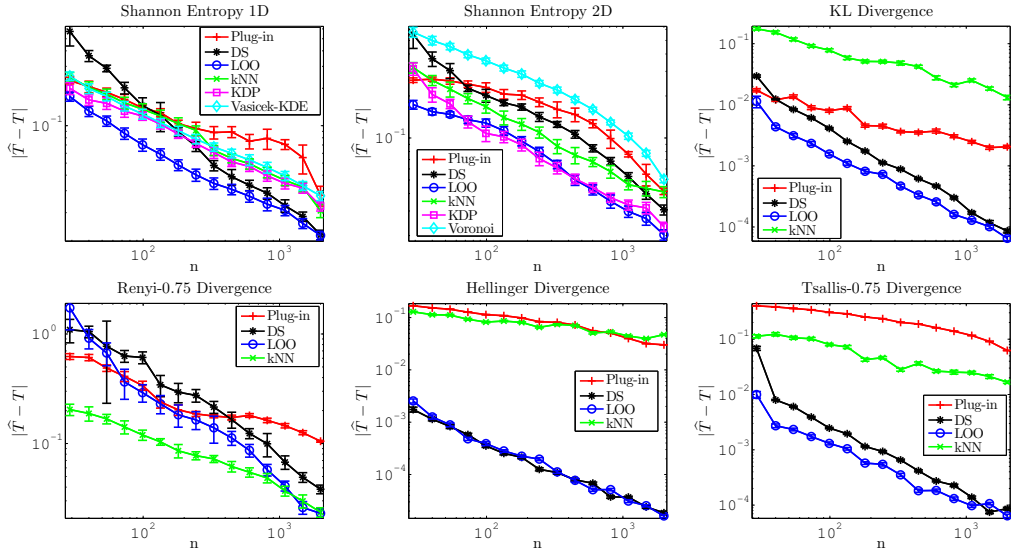

Figure 1: Comparison of DS/LOO estimators against alternatives on different functionals. The $y$-axis is the error $|\widehat{T} - T(f,g)|$ and the $x$-axis is the number of samples. All curves were produced by averaging over 50 experiments. Discretisation in hyperparameter selection may explain some of the unsmooth curves.

## 5 Comparison with Other Approaches

Estimation of statistical functionals under nonparametric assumptions has received considerable attention over the last few decades. A large body of work has focused on estimating the Shannon entropy– Beirlant et al. [1] gives a nice review of results and techniques. More recent work in the single-distribution setting includes estimation of Rényi and Tsallis entropies [17, 24]. There are also several papers extending some of these techniques to divergence estimation [10, 12, 26, 27, 37].

Many of the existing methods can be categorised as plug-in methods: they are based on estimating the densities either via a KDE or using $k$-Nearest Neighbors ($k$-NN) and evaluating the functional on these estimates. Plug-in methods are conceptually simple but unfortunately suffer several drawbacks. First, they typically have worse convergence rate than our approach, achieving the parametric rate only when $s \geq d$ as opposed to $s \geq d/2$ [19, 32]. Secondly, using either the KDE or $k$-NN, obtaining the best rates for plug-in methods requires undersmoothing the density estimate and we are not aware for principled approaches for selecting this smoothing parameter. In contrast, the bandwidth used in our estimators is the optimal bandwidth for density estimation so we can select it using a number of approaches, e.g. cross validation. This is convenient from a practitioners perspective as the bandwidth can be selected automatically, a convenience that other estimators do not enjoy. Secondly, plugin methods based on the KDE always require computationally burdensome numeric integration. In our approach, numeric integration can be avoided for many functionals of interest (See Table 1).

Another line of work focuses more specifically on estimating $f$-Divergences. Nguyen et al. [22] estimate $f$-divergences by solving a convex program and analyse the method when the likelihood ratio of the densities belongs to an RKHS. Comparing the theoretical results is not straightforward as it is not clear how to port the RKHS assumption to our setting. Further, the size of the convex program increases with the sample size which is problematic for large samples. Moon and Hero [21] use a weighted ensemble estimator for $f$-divergences. They establish asymptotic normality and the parametric convergence rate only when $s \geq d$, which is a stronger smoothness assumption than is required by our technique. Both these works only consider $f$-divergences, whereas our method has wider applicability and includes $f$-divergences as a special case.

## 6 Experiments

We compare the estimators derived using our methods on a series of synthetic examples. We compare against the methods in [8, 20, 23, 26–29, 33]. Software for the estimators was obtained either

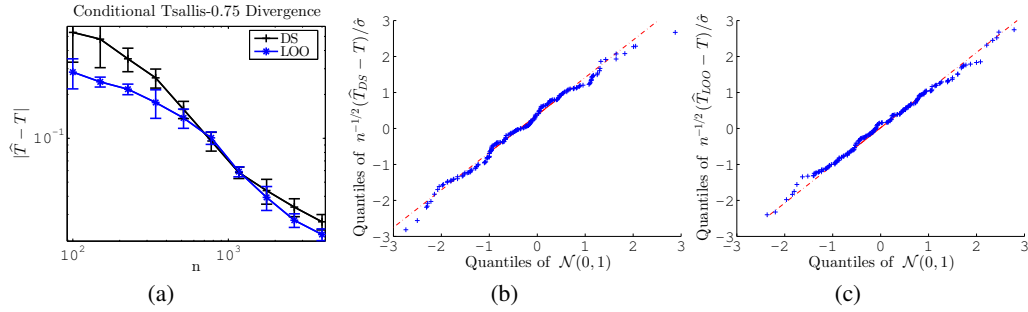

Figure 2: Fig (a): Comparison of the LOO vs DS estimator on estimating the Conditional Tsallis divergence in 4 dimensions. Note that the plug-in estimator is intractable due to numerical integration. There are no other known estimators for the conditional tsallis divergence. Figs (b), (c): QQ plots obtained using 4000 samples for Hellinger divergence estimation in 4 dimensions using the DS and LOO estimators respectively.

directly from the papers or from Szabó [34]. For the DS/LOO estimators, we estimate the density via a KDE with the smoothing kernels constructed using Legendre polynomials [35]. In both cases and for the plug in estimator we choose the bandwidth by performing 5-fold cross validation. The integration for the plug in estimator is approximated numerically.

We test the estimators on a series of synthetic datasets in $1 - 4$ dimension. The specifics of the densities used in the examples and methods compared to are given in Appendix G. The results are shown in Figures 1 and 2. We make the following observations. In most cases the LOO estimator performs best. The DS estimator approaches the LOO estimator when there are many samples but is generally inferior to the LOO estimator with few samples. This, as we have explained before is because data splitting does not make efficient use of the data. The $k$-NN estimator for divergences [28] requires choosing a $k$. For this estimator, we used the default setting for $k$ given in the software. As performance is sensitive to the choice of $k$, it performs well in some cases but poorly in other cases. We reiterate that the hyper-parameter of our estimator (bandwidth of the kernel) can be selected automatically using cross validation.

Next, we test the DS and LOO estimators for asymptotic normality on a 4-dimensional Hellinger divergence estimation problem. We use 4000 samples for estimation. We repeat this experiment 200 times and compare the empiriical asymptotic distribution (i.e. the $\sqrt{4000}(\widehat{T} - T(f, g))/\widehat{S}$ values where $\widehat{S}$ is the estimated asymptotic variance) to a $\mathcal{N}(0, 1)$ distribution on a QQ plot. The results in Figure 2 suggest that both estimators are asymptotically normal.

**Image clustering:** We demonstrate the use of our nonparametric divergence estimators in an image clustering task on the ETH-80 datset [16]. Using our Hellinger divergence estimator we achieved an accuracy of $92.47\%$ whereas a naive spectral clustering approach achieved only $70.18\%$. When we used a $k$-NN estimator for the Hellinger divergence [28] we achieved $90.04\%$ which attests to the superiority of our method. Since this is not the main focus of this work we defer this to Appendix G.

# 7   Conclusion

We generalise existing results in Von Mises estimation by proposing an empirically superior LOO technique for estimating functionals and extending the framework to functionals of two distributions. We also prove a lower bound for the latter setting. We demonstrate the practical utility of our technique via comparisons against other alternatives and an image clustering application. An open problem arising out of our work is to derive the limiting distribution of the LOO estimator.

### Acknowledgements

This work is supported in part by NSF Big Data grant IIS-1247658 and DOE grant DESC0011114.

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
