[Supplementary Material]

# Appendix

## A    Auxiliary Results

**Lemma 9** (VME and Functional Taylor Expansion). *Let $P, Q$ have densities $p, q$ and let $T(P) = \phi(\int \nu(p))$. Then the first order VME of $T(Q)$ around $P$ reduces to a functional Taylor expansion around $p$:*

$$T(Q) = T(P) + T'(Q - P; P) + R_2 = T(p) + \phi'\left(\int \nu(p)\right) \int \nu'(p)(q - p) + R_2 \qquad (12)$$

*Proof.* It is sufficient to show that the first order terms are equal.

$$T'(Q - P; P) = \frac{\partial T((1-t)P + tQ)}{\partial t}\Big|_{t=0} = \frac{\partial}{\partial t}\phi\left(\int \nu((1-t)p + tq)\right)\Big|_{t=0}$$

$$= \phi'\left(\int \nu((1-t)p + tq)\right) \int \nu'((1-t)p + tq)(q - p)\big|_{t=0}$$

$$= \phi'\left(\int \nu(p)\right) \int \nu'(p)(q - p)$$

$\square$

**Lemma 10** (VME and Functional Taylor Expansion - Two Distributions). *Let $P_1, P_2, Q_1, Q_2$ be distributions with densities $p_1, p_2, q_1, q_2$. Let $T(P_1, P_2) = \int \nu(p_1, p_2)$. Then,*

$$T(Q_1, Q_2) = T(P_1, P_2) + T_1'(Q_1 - P_1; P_1, P_2) + T_2'(Q_2 - P_2; P_1, P_2) + R_2 \qquad (13)$$

$$= T(P_1, P_2) + \phi'\left(\int \nu(p)\right) \left(\int \frac{\partial \nu(p_1(x), p_2(x))}{\partial p_1(x)}(q_1 - p_1)\mathrm{d}x + \int \frac{\partial \nu(p_1(x), p_2(x))}{\partial p_2(x)}(q_2 - p_2)\mathrm{d}x\right) + R_2$$

*Proof.* Is similar to Lemma 9.

$\square$

**Lemma 11.** *Let $f, g$ be two densities bounded above and below on a compact space. Then for all $a, b$*

$$\|f^a - g^a\|_b \in O(\|f - g\|_b)$$

*Proof.* Follows from the expansion,

$$\int |f^a - g^a|^b = \int |g^a(x) + a(f(x) - g(x))g_*^{a-1}(x) - g^a(x)|^b \leq a^b \sup |g_*^{b(a-1)}(x)| \int |f - g|^b.$$

Here $g_*(x)$ takes an intermediate value between $f(x)$ and $g(x)$. In the second step we have used the boundedness of $f$, $g$ to bound $f_*$.

$\square$

Finally, we will make use of the Efron Stein inequality stated below in our analysis.

**Lemma 12** (Efron-Stein Inequality). *Let $X_1, \ldots, X_n, X_1', \ldots, X_n'$ be independent random variables where $X_i, X_i' \in \mathcal{X}_i$. Let $Z = f(X_1, \ldots, X_n)$ and $Z^{(i)} = f(X_1, \ldots, X_i', \ldots, X_n)$ where $f : \mathcal{X}_1 \times \cdots \times \mathcal{X}_n \to \mathbb{R}$. Then,*

$$\mathbb{V}(Z) \leq \frac{1}{2}\, \mathbb{E}\left[\sum_{i=1}^n (Z - Z^{(i)})^2\right]$$

## B    Review: DS Estimator on a Single Distribution

This section is intended to be a review of the data split estimator used in [30]. The estimator was originally analysed in the semiparametric setting. However, in order to be self contained we provide an h analysis that directly uses the Von Mises Expansion. We state our main result below.

**Theorem 13.** *Suppose $f \in \Sigma(s, L, B, B')$ and $\psi$ satisfies Assumption 4. Then, $\mathbb{E}[(\widehat{T}_{\mathrm{DS}} - T(f))^2]$ is $\mathcal{O}(n^{\frac{-4s}{2s+d}})$ when $s < d/2$ and $\mathcal{O}(n^{-1})$ when $s > d/2$. Further, when $s > d/2$ and when $\psi_f, \psi_g \neq \mathbf{0}$, $\widehat{T}_{\mathrm{DS}}$ is asymptotically normal.*

$$\sqrt{n}(\widehat{T}_{\mathrm{DS}} - T(f,g)) \xrightarrow{\mathcal{D}} \mathcal{N}\left(0, \frac{1}{\zeta}\mathbb{V}_f\left[\psi_f(X; f, g)\right] + \frac{1}{1-\zeta}\mathbb{V}_g\left[\psi_g(Y; f, g)\right]\right) \tag{14}$$

We begin the proof with a series of technical lemmas.

**Lemma 14.** *The Influence Function has zero mean. i.e. $\mathbb{E}_P[\psi(X; P)] = 0$.*

*Proof.* $0 = T'(P - P; P) = \int \psi(x; P)\mathrm{d}P(x).$ $\qquad\qquad\square$

Now we prove the following lemma on the preliminary estimator $\widehat{T}_{\mathrm{DS}}^{(1)}$.

**Lemma 15** (Conditional Bias and Variance). *Let $\hat{f}^{(1)}$ be a consistent estimator for $f$ in the $L_2$ metric. Let $T$ have bounded second derivatives and let $\sup_x \psi(x; f)$ and $\mathbb{V}_{X \sim f}\psi(X; g)$ be bounded for all $g \in \mathcal{M}$. Then, the bias of the preliminary estimator $\widehat{T}_{\mathrm{DS}}^{(1)}$ (7) conditioned on $X_1^{n/2}$ is $\mathcal{O}(\|f - \hat{f}^{(1)}\|_2^2)$. The conditional variance is $\mathcal{O}(1/n)$.*

*Proof.* First consider the conditional bias,

$$\mathbb{E}_{X_{n/2+1}^n}\left[\widehat{T}_{\mathrm{DS}}^{(1)} - T(f)|X_1^{n/2}\right] = \mathbb{E}_{X_{n/2+1}^n}\left[T(\hat{f}^{(1)}) + \frac{2}{n}\sum_{i=n/2+1}^n \psi(X_i; \hat{f}^{(1)}) - T(f)|X_1^{n/2}\right]$$

$$= T(\hat{f}^{(1)}) + \mathbb{E}_f\left[\psi(X; \hat{f}^{(1)})\right] - T(f) \in \mathcal{O}(\|\hat{f}^{(1)} - f\|_2^2). \tag{15}$$

The last step follows from the boundedness of the second derivative from which the first order functional Taylor expansion (4) holds. The conditional variance is,

$$\mathbb{V}_{X_{n/2+1}^n}\left[\widehat{T}_{\mathrm{DS}}^{(1)}|X_1^{n/2}\right] = \mathbb{V}_{X_{n/2+1}^n}\left[\frac{2}{n}\sum_{i=n/2+1}^n \psi(X; \hat{f}^{(1)})\Big|X_1^{n/2}\right] = \frac{2}{n}\mathbb{V}_f\left[\psi(X; \hat{f}^{(1)})\right] \in \mathcal{O}(n^{-1}).$$

$$\tag{16}$$

$\square$

**Lemma 16** (Asymptotic Normality). *Suppose in addition to the conditions in the lemma above we also have Assumption 4 and $\|\hat{f}^{(1)} - f\|_2 \in o_P(n^{-1/4})$ and $\psi \neq \mathbf{0}$. Then,*

$$\sqrt{n}(\widehat{T}_{\mathrm{DS}} - T(f)) \xrightarrow{\mathcal{D}} \mathcal{N}(0, \mathbb{V}_f\psi(X; f)).$$

*Proof.* We begin with the following expansion around $\hat{f}^{(1)}$,

$$T(f) = T(\hat{f}^{(1)}) + \int \psi(u; \hat{f}^{(1)})f(u)\mathrm{d}\mu(u) + O(\|\hat{f}^{(1)} - f\|^2). \tag{17}$$

First consider $\widehat{T}_{\mathrm{DS}}^{(1)}$. We can write

$$\sqrt{\frac{n}{2}}\left(\widehat{T}_{\mathrm{DS}}^{(1)} - T(f)\right) = \sqrt{\frac{n}{2}}\left(T(\hat{f}^{(1)}) + \frac{2}{n}\sum_{i=n/2+1}^n \psi(X_i; \hat{f}^{(1)}) - T(f)\right) \tag{18}$$

$$= \sqrt{\frac{2}{n}} \sum_{i=n/2+1}^{n} \left[ \psi(X_i; \hat{f}^{(1)}) - \psi(X_i; f) - \left( \int \psi(u; \hat{f}^{(1)}) f(u) \mathrm{d}u - \int \psi(u; f) f(u) \mathrm{d}u \right) \right]$$

$$+ \sqrt{\frac{2}{n}} \sum_{i=n/2+1}^{n} \psi(X_i; f) + \sqrt{n} \mathcal{O} \left( \| \hat{f}^{(1)} - f \|^2 \right).$$

In the second step we used the VME in (17). In the third step, we added and subtracted $\sum_i \psi(X_i; f)$ and also added $\mathbb{E}\psi(\cdot; f) = 0$. Above, the third term is $o_P(1)$ as $\| \hat{f}^{(1)} - f \|_2 \in o_P(n^{-1/4})$. The first term which we shall denote by $Q_n$ can also be shown to be $o_P(1)$ via Chebyshev's inequality. It is sufficient to show $\mathbb{P}(|Q_n| > \epsilon | X_1^{n/2}) \to 0$. First note that,

$$\mathbb{V}[Q_n | X_1^{n/2}] = \mathbb{V} \left[ \sqrt{\frac{2}{n}} \sum_{i=n/2+1}^{n} \left( \psi(X_i; \hat{f}^{(1)}) - \psi(X_i; f) - \left( \int \psi(u; \hat{f}^{(1)}) f(u) \mathrm{d}u - \int \psi(u; f) f(u) \mathrm{d}u \right) \right) \Big| X_1^{n/2} \right]$$

$$= \mathbb{V} \left[ \psi(X; \hat{f}^{(1)}) - \psi(X; f) - \left( \int \psi(u; \hat{f}^{(1)}) f(u) \mathrm{d}u - \int \psi(u; f) f(u) \mathrm{d}u \right) \Big| X_1^{n/2} \right]$$

$$\leq \mathbb{E} \left[ \left( \psi(X; \hat{f}^{(1)}) - \psi(X; f) \right)^2 \right] \in \mathcal{O}(\| \hat{f}^{(1)} - f \|^2) \to 0, \qquad (19)$$

where the last step follows from Assumption 4. Now, $\mathbb{P}(|Q_n| > \epsilon | X_1^n) \leq \mathbb{V}(Q_n | X_1^{n/2})/\epsilon \to 0$. Hence we have

$$\sqrt{\frac{n}{2}} (\widehat{T}_{\mathrm{DS}}^{(1)} - T(f)) = \sqrt{\frac{2}{n}} \sum_{i=n/2+1}^{n} \psi(X_i; f) + o_P(1)$$

We can similarly show

$$\sqrt{\frac{n}{2}} (\widehat{T}_{\mathrm{DS}}^{(2)} - T(f)) = \sqrt{\frac{2}{n}} \sum_{i=n/2+1}^{n} \psi(X_i; f) + o_P(1)$$

Therefore, by the CLT and Slutzky's theorem,

$$\sqrt{n} (\widehat{T}_{\mathrm{DS}} - T(f)) = \frac{1}{\sqrt{2}} \left( \sqrt{\frac{n}{2}} (\widehat{T}_{\mathrm{DS}}^{(1)} - T(f)) + \sqrt{\frac{n}{2}} (\widehat{T}_{\mathrm{DS}}^{(2)} - T(f)) \right)$$

$$= n^{-1/2} \sum_{i=1}^{n} \psi(X_i; f) + o_P(1) \xrightarrow{\mathcal{D}} \mathcal{N}(0, \mathbb{V}_f \psi(X; f))$$

$\square$

We are now ready to prove Theorem 13. Note that the brunt of the work for the DS estimator was in analysing the preliminary estimator $\widehat{T}_{\mathrm{DS}}$.

*Proof of Theorem 13.* We first note that in a Hölder class, with $n$ samples the KDE achieves the rate $\mathbb{E}\| p - \hat{p} \|^2 \in O(n^{\frac{-2s}{2s+d}})$. Then the bias of $\widehat{T}_{\mathrm{DS}}$ is,

$$\mathbb{E}_{X_1^{n/2}} \mathbb{E}_{X_{n/2+1}^n} \left[ \widehat{T}_{\mathrm{DS}}^{(1)} - T(f) | X_1^{n/2} \right] = \mathbb{E}_{X_1^{n/2}} \left[ O \left( \| f - \hat{f}^{(1)} \|^2 \right) \right] \in O \left( n^{\frac{-2s}{2s+d}} \right)$$

It immediately follows that $\mathbb{E} \left[ \widehat{T}_{\mathrm{DS}} - T(f) \right] \in O \left( n^{\frac{-2s}{2s+d}} \right)$. For the variance, we use Theorem 15 and the Law of total variance for $\widehat{T}_{\mathrm{DS}}^{(1)}$,

$$\mathbb{V}_{X_1^n} \left[ \widehat{T}_{\mathrm{DS}}^{(1)} \right] = \frac{1}{n} \mathbb{E}_{X_1^{n/2}} \mathbb{V}_f \left[ \psi(X; \hat{f}^{(1)}, \hat{g}) \right] + + \mathbb{V}_{X_1^{n/2}} \left[ \mathbb{E}_{X_{n/2+1}^n} \left[ \widehat{T}_{\mathrm{DS}} - T(f) | X_1^{n/2} \right] \right]$$

$$\in O \left( \frac{1}{n} \right) + \mathbb{E}_{X_1^{n/2}} \left[ O \left( \| f - \hat{f}^{(1)} \|^4 \right) \right]$$

$$\in O \left( n^{-1} + n^{\frac{-4s}{2s+d}} \right)$$

In the second step we used the fact that $\mathbb{V}Z \leq \mathbb{E}Z^2$. Further, $\mathbb{E}_{X_1^{n/2}} \mathbb{V}_f \left[ \psi(X; \hat{f}^{(1)}) \right]$ is bounded since $\psi$ is bounded. The variance of $\widehat{T}_{\mathrm{DS}}$ can be bounded using the Cauchy Schwarz Inequality,

$$\mathbb{V}\left[\widehat{T}_{\mathrm{DS}}\right] = \mathbb{V}\left[\frac{\widehat{T}_{\mathrm{DS}}^{(1)} + \widehat{T}_{\mathrm{DS}}^{(2)}}{2}\right] = \frac{1}{4}\left(\mathbb{V}\widehat{T}_{\mathrm{DS}}^{(1)} + \mathbb{V}\widehat{T}_{\mathrm{DS}}^{(2)} + 2\mathbb{C}\mathrm{ov}(\widehat{T}_{\mathrm{DS}}^{(1)}, \widehat{T}_{\mathrm{DS}}^{(2)})\right)$$

$$\leq \max\left(\mathbb{V}\widehat{T}_{\mathrm{DS}}^{(1)}, \mathbb{V}\widehat{T}_{\mathrm{DS}}^{(2)}\right) \in O\left(n^{-1} + n^{\frac{-4s}{2s+d}}\right)$$

Finally for asymptotic normality, when $s > d/2$, $\mathbb{E}\|\hat{f}^{(1)} - f\|_2 \in \mathcal{O}(n^{\frac{-s}{2s+d}}) \in o\left(n^{-1/4}\right)$. $\quad\square$

# C  Analysis of LOO Estimator

*Proof of Theorem 5.* First note that we can bound the mean squared error via the bias and variance terms.
$$\mathbb{E}[(\widehat{T}_{\mathrm{LOO}} - T(f))^2] \leq |\mathbb{E}\widehat{T}_{\mathrm{LOO}} - T(f)|^2 + \mathbb{E}[(\widehat{T}_{\mathrm{LOO}} - \mathbb{E}\widehat{T}_{\mathrm{LOO}})^2]$$
The bias is bounded via a straightforward conditioning argument.

$$\mathbb{E}|\widehat{T}_{\mathrm{LOO}} - T(f)| = \mathbb{E}[T(\hat{f}_{-i}) + \psi(X_i; \hat{f}_{-i}) - T(f)] = \mathbb{E}_{X_{-i}}\left[\mathbb{E}_{X_i}[T(\hat{f}_{-i}) + \psi(X_i; \hat{f}_{-i}) - T(f)]\right]$$

$$= \mathbb{E}_{X_{-i}}\left[\mathcal{O}(\|\hat{f}_{-i} - f\|^2)\right] \leq C_1 n^{\frac{-2s}{2s+d}} \tag{20}$$

for some constant $C_1$. The last step follows by observing that the KDE achieves the rate $n^{\frac{-2s}{2s+d}}$ in integrated squared error.

To bound the variance we use the Efron-Stein inequality. For this consider two sets of samples $X_1^n = \{X_1, X_2, \ldots, X_n\}$ and $X_1^{n\prime} = \{X_1', X_2, \ldots, X_n\}$ which are the same except for the first point. Denote the estimators obtained using $X_1^n$ and $X_1^{n\prime}$ by $\widehat{T}_{\mathrm{LOO}}$ and $\widehat{T}_{\mathrm{LOO}}'$ respectively. To apply Efron-Stein we shall bound $\mathbb{E}[(\widehat{T}_{\mathrm{LOO}} - \widehat{T}_{\mathrm{LOO}}')^2]$. Note that,

$$|\widehat{T}_{\mathrm{LOO}} - \widehat{T}_{\mathrm{LOO}}'| \leq \frac{1}{n}|\psi(X_1; \hat{f}_{-1}) - \psi(X_1'; \hat{f}_{-1})| + \frac{1}{n}\sum_{i\neq 1}|T(\hat{f}_{-i}) - T(\hat{f}_{-i}')|$$

$$+ \frac{1}{n}\sum_{i\neq 1}|\psi(X_i; \hat{f}_{-i}) - \psi(X_i; \hat{f}_{-i}')| \tag{21}$$

The first term can be bounded by $2\|\psi\|_\infty/n$ using the boundedness of $\psi$. Each term inside the summation in the second term in (21) can be bounded via,

$$|T(\hat{f}_{-i}) - T(\hat{f}_{-i}')| \leq L_\phi \int |\nu(\hat{f}_{-i}) - \nu(\hat{f}_{-i}')| \leq L_\nu L_\nu \int |\hat{f}_{-i} - \hat{f}_{-i}'| \tag{22}$$

$$\leq L_\phi L_\nu \int \frac{1}{nh^d}\left|K\left(\frac{X_1 - u}{h}\right) - K\left(\frac{X_1' - u}{h}\right)\right|\mathrm{d}u \leq \frac{\|K\|_\infty L_\phi L_\nu}{n}.$$

The substitution $(X_1 - u)/h = z$ for integration eliminates the $1/h^d$. Here $L_\phi, L_\nu$ are the Lipschitz constants of $\phi, \nu$. To apply Efron-Stein we need to bound the expectation of the LHS over $X_1, X_1', X_2, \ldots, X_n$. Since the first two terms in (21) are bounded pointwise by $\mathcal{O}(1/n^2)$ they are also bounded in expectation. By Jensen's inequality we can write,

$$|\widehat{T}_{\mathrm{LOO}} - \widehat{T}_{\mathrm{LOO}}'|^2 \leq \frac{12\|\psi\|_\infty^2}{n^2} + \frac{3\|K\|_\infty^2 L_\phi^2 L_\nu^2}{n^2} + \frac{3}{n^2}\left(\sum_{i\neq 1}|\psi(X_i; \hat{f}_{-i}) - \psi(X_i; \hat{f}_{-i}')|\right)^2 \tag{23}$$

For the third, such a pointwise bound does not hold so we will directly bound the expectation.

$$\sum_{1\neq i,j}\mathbb{E}\left[|\psi(X_i; \hat{f}_{-i}) - \psi(X_i; \hat{f}_{-i}')||\psi(X_j; \hat{f}_{-j}) - \psi(X_j; \hat{f}_{-j}')|\right] \tag{24}$$

We then have,

$$\mathbb{E}\big[(\psi(X_i; \hat{f}_{-i}) - \psi(X_i; \hat{f}'_{-i}))^2\big] \le \mathbb{E}_{X_1, X_1'}\left[C \int |\hat{f}_{-i} - \hat{f}'_{-i}|^2\right]$$

$$\le CB^2 \int \frac{1}{n^2 h^{2d}}\left(K\left(\frac{x_1 - u}{h}\right) - K\left(\frac{x_1' - u}{h}\right)\right)^2 \mathrm{d}x_1 \mathrm{d}x_1' u$$

$$\le \frac{2CB^2\|K\|_\infty^2}{n^2}$$

I the first step we have used Assumption 4 and in the last step the substitutions $(x_1 - x_i)/h = u$ and $(x_1 - x_j)/h = v$ removes the $1/h^d$ twice. Then, by applying Cauchy Schwarz each term inside the summation in (24) is $\mathcal{O}(1/n^2)$.

Since each term inside equation (24) is $\mathcal{O}(1/n^2)$ and there are $(n-1)^2$ terms it is $\mathcal{O}(1)$. Combining all these results with equation (23) we get,

$$\mathbb{E}[(\widehat{T}_{\mathrm{LOO}} - \widehat{T}'_{\mathrm{LOO}})^2] \in \mathcal{O}\left(\frac{1}{n^2}\right)$$

Now, by applying the Efron-Stein inequality we get $\mathbb{V}(\widehat{T}_{\mathrm{LOO}}) \le \frac{C}{2n}$. Therefore the mean squared error $\mathbb{E}[(T - \widehat{T}_{\mathrm{LOO}})^2] \in \mathcal{O}(n^{-\frac{4s}{2s+d}} + n^{-1})$ which completes the proof. □

# D  Proofs of Results on Functionals of Two Distributions

## D.1  DS Estimator

We generalise the results in Appendix B to analyse the DS estimator for two distributions. As before we begin with a series of lemmas.

**Lemma 17.** *The influence functions have zero mean. I.e.*

$$\mathbb{E}_{P_1}[\psi_1(X; P_1; P_2)] = 0 \quad \forall P_2 \in \mathcal{M} \qquad \mathbb{E}_{P_2}[\psi_2(Y; P_1; P_2)] = 0 \quad \forall P_1 \in \mathcal{M} \qquad (25)$$

*Proof.* $0 = T_i'(P_i - P_i; P_1; P_2) = \int \psi_i(u; P_1, P_2)\mathrm{d}P_i(u)$ for $i = 1, 2$. □

**Lemma 18** (Bias & Variance of (9)). *Let $\hat{f}^{(1)}, \hat{g}^{(1)}$ be consistent estimators for $f, g$ in the $L_2$ sense. Let $T$ have bounded second derivatives and let $\sup_x \psi_f(x; f, g)$, $\sup_x \psi_g(x; f, g)$, $\mathbb{V}_f\psi(X; f', g')$, $\mathbb{V}_g\psi_g(X; f', g')$ be bounded for all $f, f', g, g' \in \mathcal{M}$. Then the bias of $\widehat{T}_{\mathrm{DS}}^{(1)}$ conditioned on $X_1^{n/2}, Y_1^{m/2}$ is $|T - \mathbb{E}[\widehat{T}_{\mathrm{DS}}^{(1)}|X_1^{n/2}, Y_1^{m/2}] \in \mathcal{O}(\|f - \hat{f}^{(1)}\|^2 + \|g - \hat{g}^{(1)}\|^2)$. The conditional variance is $\mathbb{V}[\widehat{T}_{\mathrm{DS}}^{(1)}|X_1^{n/2}, Y_1^{m/2}] \in \mathcal{O}(n^{-1} + m^{-1})$.*

*Proof.* First consider the bias conditioned on $X_1^{n/2}, Y_1^{m/2}$,

$$\mathbb{E}\left[\widehat{T}_{\mathrm{DS}}^{(1)} - T(f, g)|X_1^{n/2}, Y_1^{m/2}\right]$$

$$= \mathbb{E}\left[T(\hat{f}^{(1)}, \hat{g}^{(1)}) + \frac{2}{n}\sum_{i=n/2+1}^n \psi_f(X_i; \hat{f}^{(1)}, \hat{g}^{(1)}) + \frac{2}{m}\sum_{j=m/2+1}^m \psi_g(Y_j; \hat{f}^{(1)}, \hat{g}^{(1)}) - T(f, g)\bigg|X_1^{n/2}, Y_1^{m/2}\right]$$

$$= T(\hat{f}^{(1)}, \hat{g}^{(1)}) + \int \psi_f(x; \hat{f}^{(1)}, \hat{g}^{(1)})f(x)\mathrm{d}\mu(x) + \int \psi_g(x; \hat{f}^{(1)}, \hat{g}^{(1)})g(x)\mathrm{d}\mu(x) - T(f, g)$$

$$= \mathcal{O}\left(\|f - \hat{f}^{(1)}\|^2 + \|g - \hat{g}^{(1)}\|^2\right)$$

The last step follows from the boundedness of the second derivatives from which the first order functional Taylor expansion (6) holds. The conditional variance is,

$$\mathbb{V}\left[\widehat{T}_{\mathrm{DS}}^{(1)}|X_1^{n/2}, Y_1^{m/2}\right] = \mathbb{V}\left[\frac{1}{n}\sum_{i=n+1}^{2n}\psi_f(X_i; \hat{f}^{(1)}, \hat{g}^{(1)})\Big|X_1^{n/2}\right] + \mathbb{V}\left[\frac{1}{m}\sum_{j=m+1}^{2m}\psi_g(Y_j; \hat{f}^{(1)}, \hat{g}^{(1)})\Big|Y_1^{m/2}\right]$$

$$= \frac{1}{n}\mathbb{V}_f\left[\psi_f(X; \hat{f}^{(1)}, \hat{g}^{(1)})\right] + \frac{1}{m}\mathbb{V}_g\left[\psi_g(Y; \hat{f}^{(1)}, \hat{g}^{(1)})\right] \in \mathcal{O}\left(\frac{1}{n} + \frac{1}{m}\right)$$

The last step follows from the boundedness of the variance of the influence functions. $\qquad\square$

The following lemma characterises conditions for asymptotic normality.

**Lemma 19** (Asymptotic Normality). *Suppose, in addition to the conditions in Theorem 18 above and the regularity assumption 4 we also have $\|\hat{f} - f\| \in o_P(n^{-1/4}), \|\hat{g} - g\| \in o_P(m^{-1/4})$ and $\psi_f, \psi_g \neq \mathbf{0}$. Then we have asymptotic Normality for $\widehat{T}_{\mathrm{DS}}$,*

$$\sqrt{N}(\widehat{T}_{\mathrm{DS}} - T(f,g)) \xrightarrow{\mathcal{D}} \mathcal{N}\left(0, \frac{1}{\zeta}\mathbb{V}_f\left[\psi_f(X; f,g)\right] + \frac{1}{1-\zeta}\mathbb{V}_g\left[\psi_g(Y; f,g)\right]\right) \qquad (26)$$

*Proof.* We begin with the following expansions around $(\hat{f}^{(1)}, \hat{g}^{(1)})$,

$$T(f,g) = T(\hat{f}^{(1)}, \hat{g}^{(1)}) + \int \psi_f(u; \hat{f}^{(1)}, \hat{g}^{(1)})f(u)\mathrm{d}u + \int \psi_g(u; \hat{f}^{(1)}, \hat{g}^{(1)})g(u)\mathrm{d}u +$$

$$\mathcal{O}\left(\|f - \hat{f}^{(1)}\|^2 + \|g - \hat{g}^{(1)}\|^2\right)$$

Consider $\widehat{T}_{\mathrm{DS}}^{(1)}$. We can write

$$\sqrt{\frac{N}{2}}(\widehat{T}_{\mathrm{DS}}^{(1)} - T(f)) \qquad\qquad (27)$$

$$= \sqrt{\frac{N}{2}}\left(T(\hat{f}^{(1)}, \hat{g}^{(1)}) + \frac{2}{n}\sum_{i=n/2+1}^{n}\psi_f(X_i; f,g) + \frac{2}{m}\sum_{j=m/2+1}^{m}\psi_g(Y_j; f,g) - T(f,g)\right)$$

$$= \sqrt{\frac{N}{2}}\left(\frac{2}{n}\sum_{i=n/2+1}^{n}\psi(X_i; \hat{f}^{(1)}, \hat{g}^{(1)}) + \frac{2}{m}\sum_{j=m/2+1}^{m}\psi(X_j; \hat{f}^{(1)}, \hat{g}^{(1)}) - \mathbb{E}_f\left[\psi(X; \hat{f}^{(1)}, \hat{g}^{(1)})\right]\right.$$

$$\left. - \mathbb{E}_g\left[\psi(X; \hat{f}^{(1)}, \hat{g}^{(1)})\right]\right) + \sqrt{N}O\left(\|f - \hat{f}^{(1)}\|^2 + \|g - \hat{g}^{(1)}\|^2\right)$$

$$= \sqrt{\frac{2N}{n}}n^{-1/2}\sum_{i=n/2+1}^{n}\left(\psi_f(X_i; \hat{f}^{(1)}, \hat{g}^{(1)}) - \psi_f(X_i; f,g) - (\mathbb{E}_f\psi_f(X; \hat{f}^{(1)}, \hat{g}^{(1)}) + \mathbb{E}_f\psi_f(X; f,g))\right) +$$

$$\sqrt{\frac{2N}{m}}m^{-1/2}\sum_{j=m/2+1}^{m}\left(\psi_g(Y_j; \hat{f}^{(1)}, \hat{g}^{(1)}) - \psi_g(Y_j; f,g) - (\mathbb{E}_g\psi_g(Y; \hat{f}^{(1)}, \hat{g}^{(1)}) + \mathbb{E}_g\psi_g(Y; f,g))\right) +$$

$$\sqrt{\frac{2N}{n}}n^{-1/2}\sum_{i=n/2+1}^{n}\psi_f(X_i; f,g) + \sqrt{\frac{2N}{m}}m^{-1/2}\sum_{j=m/2+1}^{m}\psi_g(Y_j; f,g) +$$

$$\sqrt{N}\mathcal{O}\left(\|f - \hat{f}^{(1)}\|^2 + \|g - \hat{g}^{(1)}\|^2\right)$$

The fifth term is $o_P(1)$ by the assumptions. The first and second terms are also $o_P(1)$. To see this, denote the first term by $Q_n$.

$$\mathbb{V}\left[Q_n|X_1^{n/2}, Y_1^{m/2}\right] = \frac{N}{n}\mathbb{V}_f\left[\sum_{i=n/2+1}^{n}\left(\psi_f(X; \hat{f}^{(1)}, \hat{g}^{(1)}) - \psi_f(X; f,g) - (\mathbb{E}_f\psi_f(X; \hat{f}^{(1)}, \hat{g}^{(1)}) + \mathbb{E}_f\psi_f(X; f,g))\right)\right]$$

$$\leq \frac{N}{n}\mathbb{E}_f\left[\left(\psi_f(X_i; \hat{f}^{(1)}, \hat{g}^{(1)}) - \psi_f(X_i; f,g)\right)^2\right] \to 0$$

where we have used the regularity assumption 4. Further, $\mathbb{P}(|Q_n| > \epsilon | X_1^{n/2}, Y_1^{m/2}) \leq \mathbb{V}[Q_n|X_1^{n/2}, Y_1^{m/2}]\,\epsilon \to 0$, hence the first term is $o_P(1)$. The proof for the second term is similar. Therefore we have,

$$\sqrt{\frac{N}{2}}(\widehat{T}_{\mathrm{DS}}^{(1)} - T(f)) = \sqrt{\frac{2N}{n}}n^{-1/2}\sum_{i=n/2+1}^{n}\psi_f(X_i; f, g) + \sqrt{\frac{2N}{m}}m^{-1/2}\sum_{j=m/2+1}^{m}\psi_g(Y_j; f, g) + o_P(1)$$

Using a similar argument on $\widehat{T}_{\mathrm{DS}}^{(2)}$ we get,

$$\sqrt{\frac{N}{2}}(\widehat{T}_{\mathrm{DS}}^{(2)} - T(f)) = \sqrt{\frac{2N}{n}}n^{-1/2}\sum_{i=1}^{n/2}\psi_f(X_i; f, g) + \sqrt{\frac{2N}{m}}m^{-1/2}\sum_{j=1}^{m/2}\psi_g(Y_j; f, g) + o_P(1)$$

Therefore,

$$\sqrt{N}(\widehat{T}_{\mathrm{DS}}^{(2)} - T(f)) = \sqrt{2}\left(\sqrt{\frac{2N}{n}}n^{-1/2}\sum_{i=1}^{n}\psi_f(X_i; f, g) + \sqrt{\frac{2N}{m}}m^{-1/2}\sum_{j=1}^{m}\psi_g(Y_j; f, g)\right) + o_P(1)$$

$$= \sqrt{\frac{N}{n}}n^{-1/2}\sum_{i=1}^{2n}\psi_f(X_i; f, g) + \sqrt{\frac{N}{m}}m^{-1/2}\sum_{j=1}^{2m}\psi_g(Y_j; f, g) + o_P(1)$$

By the CLT and Slutzky's theorem this converges weakly to the RHS of (26). □

We are now ready to prove the rates of convergence for the DS estimator in the Hölder class.

*Proof of Theorem 13.* . We first note that in a Hölder class, with $n$ samples the KDE achieves the rate $\mathbb{E}\|p - \hat{p}\|^2 \in O(n^{\frac{-2s}{2s+d}})$. Then the bias for the preliminary estimator $\widehat{T}_{\mathrm{DS}}^{(1)}$ is,

$$\mathbb{E}\left[\widehat{T}_{\mathrm{DS}}^{(1)} - T(f, g)|X_1^{n/2}, Y_1^{m/2}\right] = \mathbb{E}_{X_1^{n/2}, Y_1^{m/2}}\left[O\left(\|f - \hat{f}^{(1)}\|^2 + \|g - \hat{g}^{(1)}\|^2\right)\right]$$
$$\in O\left(n^{\frac{-2s}{2s+d}} + m^{\frac{-2s}{2s+d}}\right)$$

The same could be said about $\widehat{T}_{\mathrm{DS}}^{(2)}$. It therefore follows that

$$\mathbb{E}\left[\widehat{T}_{\mathrm{DS}} - T\right] = \mathbb{E}\left[\frac{1}{2}\left(\widehat{T}_{\mathrm{DS}}^{(1)} - T(f)\right) + \frac{1}{2}\left(\widehat{T}_{\mathrm{DS}}^{(2)} - T(f)\right)\right] \in O\left(n^{\frac{-2s}{2s+d}} + m^{\frac{-2s}{2s+d}}\right)$$

For the variance, we use Theorem 18 and the Law of total variance to first control $\mathbb{V}\widehat{T}_{\mathrm{DS}}^{(1)}$,

$$\mathbb{V}\left[\widehat{T}_{\mathrm{DS}}^{(1)}\right] = \frac{1}{n}\mathbb{E}\left[\mathbb{V}_f\left[\psi_f(X; \hat{f}^{(1)}, \hat{g}^{(1)})|X_1^{n/2}\right]\right] + \frac{1}{m}\mathbb{E}\left[\mathbb{V}_g\left[\psi_g(Y; \hat{f}^{(1)}, \hat{g}^{(1)})|Y_1^{m/2}\right]\right]$$
$$+ \mathbb{V}\left[\mathbb{E}\left[\widehat{T}_{\mathrm{LOO}} - T(f, g)|X_1^{n/2}Y_1^{m/2}\right]\right]$$
$$\in O\left(\frac{1}{n} + \frac{1}{m}\right) + \mathbb{E}\left[O\left(\|f - \hat{f}^{(1)}\|^4 + \|g - \hat{g}^{(1)}\|^4\right)\right]$$
$$\in O\left(n^{-1} + m^{-1} + n^{\frac{-4s}{2s+d}} + m^{\frac{-4s}{2s+d}}\right)$$

In the second step we used the fact that $\mathbb{V}Z \leq \mathbb{E}Z^2$. Further, $\mathbb{E}_{X_1^{n/2}}\mathbb{V}_f\left[\psi_f(X; \hat{f}^{(1)}, \hat{g}^{(1)})\right]$, $\mathbb{E}_{Y_1^{m/2}}\mathbb{V}_g\left[\psi_g(Y; \hat{f}^{(1)}, \hat{g}^{(1)})\right]$ are bounded since $\psi_f, \psi_g$ are bounded. Then by applying the Cauchy Schwarz inequality as before we get $\mathbb{V}\widehat{T}_{\mathrm{DS}} \in O\left(n^{-1} + m^{-1} + n^{\frac{-4s}{2s+d}} + m^{\frac{-4s}{2s+d}}\right)$.

Finally when $s > d/2$, we have the required $o_P(n^{-1/4}), o_P(m^{-1/4})$ rates on $\|\hat{f} - f\|$ and $\|\hat{g} - g\|$ which gives us asymptotic normality. □

## D.2 LOO Estimator

*Proof of Theorem 7.* Assume w.l.o.g that $n > m$. As before, the bias follows via conditioning.

$$\mathbb{E}|\widehat{T}_{\text{LOO}} - T(f,g)| = \mathbb{E}[T(\hat{f}_{-i}, \hat{g}_{-i}) + \psi_f(X_i; \hat{f}_{-i}, \hat{g}_{-i}) + \psi_g(Y_i; \hat{f}_{-i}, \hat{g}_{-i}) - T(f,g)]$$
$$= \mathbb{E}\left[\mathcal{O}(\|\hat{f}_{-i} - f\|^2 + \|\hat{g} - g\|^2)\right] \leq C_1(n^{\frac{-2s}{2s+d}} + m^{\frac{-2s}{2s+d}})$$

for some constant $C_1$.

To bound the variance we use the Efron-Stein inequality. Consider the samples $\{X_1, \ldots, X_n, Y_1, \ldots, Y_m\}$ and $\{X'_1, \ldots, X_n, Y_1, \ldots, Y_m\}$ and denote the estimates obtained by $\widehat{T}_{\text{LOO}}$ and $\widehat{T}'_{\text{LOO}}$ respectively. Recall that we need to bound $\mathbb{E}[(\widehat{T}_{\text{LOO}} - \widehat{T}'_{\text{LOO}})^2]$. Note that,

$$|\widehat{T}_{\text{LOO}} - \widehat{T}'_{\text{LOO}}| \leq \frac{1}{n}|\psi_f(X_1; \hat{f}_{-1}, \hat{g}_{-1}) - \psi_f(X'_1; \hat{f}_{-1}, \hat{g}_{-1})| +$$

$$\frac{1}{n}\sum_{i \neq 1}|T(\hat{f}_{-i}, \hat{g}_{-i}) - T(\hat{f}'_{-i}, \hat{g}_{-i})| + |\psi_f(X_i; \hat{f}_{-i}, \hat{g}_{-i}) - \psi_f(X_i; \hat{f}'_{-i}, \hat{g}_{-i})| + |\psi_g(Y_i; \hat{f}_{-i}, \hat{g}_{-i}) - \psi_g(Y_i; \hat{f}'_{-i}, \hat{g}_{-i})|$$

The first term can be bounded by $2\|\psi_f\|_\infty/n$ using the boundedness of the influence function on bounded densities. By using an argument similar to Equation (22) in the one distribution case, we can also bound each term inside the summation of the second term via,

$$|T(\hat{f}_{-i}, \hat{g}_{-i}) - T(\hat{f}'_{-i}, \hat{g}_{-i})| \leq \frac{\|K\|_\infty L_\phi L_\nu}{n}$$

Then, by Jensen's inequality we have,

$$|\widehat{T}_{\text{LOO}} - \widehat{T}'_{\text{LOO}}|^2 \leq \frac{8\|\psi_f\|_\infty^2}{n^2} + \frac{4\|K\|_\infty^2 L_\phi^2 L_\nu^2}{n^2} + \frac{4}{n^2}\left(\sum_{i \neq 1}|\psi_f(X_i; \hat{f}_{-i}, \hat{g}_{-i}) - \psi_f(X_i; \hat{f}'_{-i}, \hat{g}_{-i})|\right)^2$$

$$+ \frac{4}{n^2}\left(\sum_{i \neq 1}|\psi_g(Y_i; \hat{f}_{-i}, \hat{g}_{-i}) - \psi_g(Y_i; \hat{f}'_{-i}, \hat{g}_{-i})|\right)^2$$

The third and fourth terms can be bound in expectation using a similar technique to bound the third term in equation 22. Precisely, by using Assumption (4) and Cauchy Schwarz we get,

$$\mathbb{E}\left[|\psi_f(X_i; \hat{f}_{-i}, \hat{g}_{-i}) - \psi_f(X_i; \hat{f}'_{-i}, \hat{g}_{-i})||\psi_f(X_j; \hat{f}_{-j}, \hat{g}_{-j}) - \psi_f(X_j; \hat{f}'_{-j}, \hat{g}_{-j})|\right] \leq \frac{2CB^2\|K\|_\infty^2}{n^2}$$

$$\mathbb{E}\left[|\psi_g(Y_i; \hat{f}_{-i}, \hat{g}_{-i}) - \psi_g(Y_i; \hat{f}'_{-i}, \hat{g}_{-i})||\psi_g(Y_j; \hat{f}_{-j}, \hat{g}_{-j}) - \psi_g(Y_j; \hat{f}'_{-j}, \hat{g}_{-j})|\right] \leq \frac{2CB^2\|K\|_\infty^2}{n^2}$$

This leads us to a $\mathcal{O}(1/n^2)$ bound for $\mathbb{E}[(\widehat{T}_{\text{LOO}} - \widehat{T}'_{\text{LOO}})^2]$,

$$\mathbb{E}[(\widehat{T}_{\text{LOO}} - \widehat{T}'_{\text{LOO}})^2] \leq \frac{8\|\psi_f\|_\infty^2 + 4\|K\|_\infty^2 L_\phi^2 L_\nu^2 + 16CB^2\|K\|_\infty^2}{n^2}$$

Now consider, the set of samples $\{X_1, \ldots, X_n, Y_1, \ldots, Y_m\}$ and $\{X_1, \ldots, X_n, Y'_1, \ldots, Y_m\}$ and denote the estimates obtained by $\widehat{T}_{\text{LOO}}$ and $\widehat{T}'_{\text{LOO}}$ respectively. Note that some of the $Y$ instances are repeated but each point occurs at most $n/m$ times. The remaining argument is exactly the same except that we need to account for this repetition. We have,

$$|\widehat{T}_{\text{LOO}} - \widehat{T}'_{\text{LOO}}| \leq \frac{n}{m}\frac{1}{n}|\psi_f(X_1; \hat{f}_{-1}, \hat{g}_{-1}) - \psi_f(X'_1; \hat{f}_{-1}, \hat{g}_{-1})| + \frac{n}{m}\frac{1}{n}\sum_{i \neq 1}\Big(|T(\hat{f}_{-i}, \hat{g}_{-i}) - T(\hat{f}_{-i}, \hat{g}_{-i})| +$$

$$|\psi_f(X_i; \hat{f}_{-i}, \hat{g}_{-i}) - \psi_f(X_i; \hat{f}'_{-i}, \hat{g}_{-i})| + |\psi_g(Y_i; \hat{f}_{-i}, \hat{g}_{-i}) - \psi_g(Y_i; \hat{f}'_{-i}, \hat{g}_{-i})|\Big) \qquad (28)$$

And hence,

$$\mathbb{E}[(\widehat{T}_{\text{LOO}} - \widehat{T}'_{\text{LOO}})^2] \leq \frac{\|\psi_g\|_\infty^2}{m^2} + \frac{n^2}{m^4}4\|K\|_\infty^2 L_\phi^2 L_\nu^2 + \mathcal{O}\left(\frac{n^4}{m^6}\right)$$

where the last two terms of (28) are bounded by $\mathcal{O}(n^4/m^6)$ after squaring and then taking the expectation. We have been a bit sloppy by bounding the difference by $n/m$ and not $\lceil n/m \rceil$ but it is clear that this doesn't affect the rate.

Finally by the Efron Stein inequality we have

$$\mathbb{V}(\widehat{T}_{\text{LOO}}) \in \mathcal{O}\left(\frac{1}{n} + \frac{n^4}{m^5}\right)$$

which is $\mathcal{O}(1/n + 1/m)$ if $n$ and $m$ are of the same order. This is the case if for instance there exists $\zeta_l, \zeta_u \in (0,1)$ such that $\zeta_l \leq n/m \leq \zeta_u$.

Therefore the mean squared error is $\mathbb{E}[(T - \widehat{T}_{\text{LOO}})^2] \in \mathcal{O}(n^{-\frac{4s}{2s+d}} + m^{-\frac{4s}{2s+d}} + n^{-1} + m^{-1})$ which completes the proof. $\qquad\square$

# E   Proof of Lower Bound (Theorem 8)

We will prove the lower bound in the bounded Hölder class $\Sigma(s, L, B, B')$ noting that the lower bound also applies to $\Sigma(s, L)$. Our main tool will be LeCam's method where we reduce the estimation problem to a testing problem. In the testing problem we construct a set of alternatives satisfying certain separation properties from the null. For this we will use some technical results from Birgé and Massart [3] and [12]. First we state LeCam's method below adapted to our setting. We define the squared Hellinger Divergence between two distributions $P, Q$ with densities $p, q$ to be

$$H^2(P, Q) = \int \left(\sqrt{p(x)} - \sqrt{q(x)}\right)^2 dx = 2 - 2\int p(x)q(x)dx$$

**Theorem 20.** *Let $T : \mathcal{M} \times \mathcal{M} \to \mathbb{R}$. Consider a parameter space $\Theta \subset \mathcal{M} \times \mathcal{M}$ such that $(f, g) \in \Theta$ and $(p_\lambda, q_\lambda) \in \Theta$ for all $\lambda$ in some index set $\Lambda$. Denote the distributions of $f, g, p_\lambda, q_\lambda$ by $F, G, P_\lambda, Q_\lambda$ respectively. Define $\overline{P \times Q} = \frac{1}{|\Lambda|}\sum_{\lambda \in \Lambda} P_\lambda^n \times Q_\lambda^m$. If, there exists $(f, g) \in \Theta$, $\gamma < 2$ and $\beta > 0$ such that the following two conditions are satisfied*

$$H^2(F^n \times G^m, \overline{P \times Q}) \leq \gamma$$
$$T(p_\lambda, q_\lambda) \geq T(f, g) + 2\beta \quad \forall \lambda \in \Lambda$$

*then,*

$$\inf_{\widehat{T}} \sup_{(f,g)\in\Theta} \mathbb{P}\left(|\widehat{T} - T(f,g)| > \beta\right) \frac{1}{2}\left(1 - \sqrt{\gamma(1 - \gamma/4)}\right) > 0.$$

*Proof.* The proof is a straightforward modification of Theorem 2.2 of Tsybakov [35] which we provide here for completeness.

Let $\Theta_0 = \{(p, q) \in \Theta | T(p, q) \leq T(f, g)\}$ and $\Theta_1 = \{(p, q) \in \Theta | T(p, q) \geq T(f, g) + 2\beta\}$. Hence $(f, g) \in \Theta_0$ and $(p_\lambda, q_\lambda) \in \Theta_1$ for all $\lambda \in \Lambda$. Given $n$ samples from $p'$ and $m$ samples from $q'$ consider the simple vs simple hypothesis testing problem of $H_0 : (p', q') \in \Theta_0$ vs $H_1 : (p', q') \in \Theta_1$. The probability of error $p_e$ of any test $\Psi$ test is lower bounded by

$$p_e \geq \frac{1}{2}\left(1 - \sqrt{H^2(F^n \times G^m, \overline{P \times Q})(1 - H^2(F^n \times G^m, \overline{P \times Q}))/4}\right).$$

See Lemma 2.1, Lemma 2.3 and Theorem 2.2 of Tsybakov [35]. Therefore,

$$\inf_{\psi} \sup_{(p',q')\in\Theta_0,(p'',q'')\in\Theta_0} p_e \geq \frac{1}{2}\left(1 - \sqrt{\gamma(1 - \gamma/4)}\right)$$

If we make an error in the testing problem the error in estimation is least $\beta$ in the estimation problem which completes the proof of the theorem. $\qquad\square$

Consider the set $\Gamma = \{-1, 1\}^\ell$ and a set of densities $p_\gamma = f(1 + \sum_{j=1}^\ell \gamma_j v_j)$ indexed by each $\gamma \in \Gamma$. Here $f$ is itself a density and the $v_j$'s are perturbations on $f$. We will also use the following result from Birgé and Massart [3] which bounds the Hellinger divergence between the product distribution $F^n$ and the mixture product distribution $\overline{P^n} = \frac{1}{|\Gamma|}\sum_{\gamma\in\Gamma} P_\gamma^n$.

**Proposition 21.** *Let $\{R_1, \ldots, R_\ell\}$ be a partition of $[0, 1]^d$. Let $\rho_j$ is zero except on $R_j$ and satisfies $\|\rho_j\|_\infty \leq 1$, $\int \rho_j f = 0$ and $\int \rho_j^2 f = \alpha_j$. Further, denote $\alpha = \sum_j \|\rho_j\|_\infty$, $s = n\alpha^2 \sup_j P(R_j)$ and $c = n \sup_j \alpha_j$. Then,*

$$H^2(F^n, \overline{P^n}) \leq \frac{n^2}{3} \sum_{j=1}^{\ell} \alpha_j^2.$$

We also use the following technical result from Krishnamurthy et al. [12] and adapt it to our setting.

**Proposition 22** (Taken from [12]). *Let $R_1, \ldots, R_\ell$ be a partition of $[0, 1]^d$ each having size $\ell^{-1/d}$. There exists functions $u_1, \ldots, u_\ell$ such that,*

$$\operatorname{supp}(u_j) \subset \{x | B(x, \epsilon) \subset R_j\}, \qquad \int u_j^2 \in \Theta(\ell^{-1}), \qquad \int u_j = 0,$$

$$\int \psi_f(x; f, g) u_j(x) = \int \psi_g(x; f, g) u_j(x) = 0, \qquad \|D^r u_j\|_\infty \leq \ell^{r/d} \ \forall r \ s.t \sum_j r_j \leq s + 1$$

*where $B(x, \epsilon)$ denotes an $L_2$ ball around $x$ with radius $\epsilon$. Here $\epsilon$ is any number between 0 and 1.*

*Proof.* For this we use an orthonormal system of $q \ (> 4)$ functions on $(0, 1)^d$ satisfying $\phi_1 = 1$, $\operatorname{supp}(\phi_j) \subset [\epsilon, 1 - \epsilon]^d$ for any $\epsilon > 0$ and $\|D^r\phi_j\|_\infty \leq J$ for some $J < \infty$. Now for any given functions $\eta_1, \eta_2$ we can find a function $\upsilon$ such that $\upsilon \in \operatorname{span}(\{\phi_j\})$, $\int \upsilon\phi_1 = \int \upsilon\eta_1 = \int \upsilon\eta_2 = 0$. Write $\upsilon = \sum_i c_j \phi_j$. Then $D^r\upsilon = \sum_j c_j D^r\phi_j$ which implies $\|D^r\upsilon\|_\infty \leq K\sqrt{q}$. Let $\nu(\cdot) = \frac{1}{J\sqrt{q}}\upsilon(\cdot)$. Clearly, $\int \nu^2$ is upper and lower bounded and $\|D^r\nu\|_\infty \leq 1$.

To construct the functions $u_j$, we map $(0, 1)^d$ to $R_j$ by appropriately scaling it. Then, $u_j(x) = \nu(m^{1/d}(x - \mathbf{j}))$ where $\mathbf{j}$ is the point corresponding to $\mathbf{0}$ after mapping. Moreover let $\eta_1$ be $\psi_f(\cdot; f, g)$ constrained to $R_j$ (and scaled back to fit $(0, 1)^d$). Let $\eta_2$ be the same with $\psi_g$. Now, $\int_{R_j} u_j^2 = \frac{1}{\ell} \int \nu^2 \in \Theta(\ell^{-1})$. Also, clearly $\|D^r u_j\| \leq m^{r/d}$. All 5 conditions above are satisfied. $\qquad\square$

We now have all necessary ingredients to prove the lower bound.

*Proof of Theorem 8.* To apply Theorem 20 we will need to construct the set of alternatives $\Lambda$ which contains tuples $(p_\lambda, q_\lambda)$ that satisfy the conditions of Theorem 20. First apply Proposition 22 with $\ell = \ell_1$ to obtain the index set $\tilde{\Gamma} = \{-1, 1\}^{\ell_1}$ and the functions $u_1, \ldots, u_{\ell_1}$. Apply it again with $\ell = \ell_2$ to obtain the index set $\Delta = \{-1, 1\}^{\ell_2}$ and the functions $v_1, \ldots, v_{\ell_2}$. Define $\Gamma, \Delta$ be the following set of functions which are perturbed around $f$ and $g$ respectively,

$$\Gamma = \left\{ p_\gamma = f + K_1 \sum_{j=1}^{\ell_1} \gamma_j u_j \middle| \gamma \in \tilde{\Gamma} \right\}$$

$$\Delta = \left\{ q_\delta = g + K_2 \sum_{j=1}^{\ell_2} \delta_j v_j \middle| \delta \in \tilde{\Delta} \right\}$$

Since the perturbations in Proposition 22 are condensed into the small $R_j$'s it invariably violates the Hölder assumption. The scaling $K_1$ and $K_2$ are necessary to shrink the perturbation and ensure that $p_\gamma, q_\delta \in \Sigma(s, L)$. By following essentially an identical argument to [12] (Section E.2) we have that $p_\gamma \in \Sigma(s, L)$ if $K \asymp \ell_1^{-s/d}$ and $q_\delta \in \Sigma(s, L)$ if $K_2 \asymp \ell_2^{-s/d}$. We will set $\ell_1$ and $\ell_2$ later on to obtain the required rates. For future reference denote $\overline{P^n} = \frac{1}{|\Gamma|} \sum_{\gamma \in \Gamma} P_\gamma^n$ and $\overline{Q^m} = \left(\frac{1}{|\Delta|} \sum_{\delta \in \Delta} Q_\delta^m\right)$.

Now our set of alternatives are formed by the product of $\Gamma$ and $\Delta$

$$\Lambda = \Gamma \times \Delta = \{(p_\gamma, q_\delta) | p_\gamma \in \Gamma, q_\delta \in \Delta\}$$

First note that for any $(p_\lambda, q_\lambda) = (p_\gamma, q_\delta) \in \Lambda$, by the second order functional Taylor expansion we have,

$$T(p_\lambda, q_\lambda) = T(f, g) + \int \psi_f(x; f, g)p_\lambda + \int \psi_g(x; f, g)q_\lambda + R_2$$

By Lemma 17 and the construction the first order terms vanish since,

$$\int \psi_f(x; f, g) \left( f + K_1 \sum_j \gamma_j u_j \right) = K_1 \sum_j \gamma_j \int \psi_f(x; f, g) u_j = 0.$$

The same is true for $\int \psi_g(x; f, g)$. The second order term can be upper bounded by

$$R_2 = \phi'' \left( \int \nu(f^*, g^*) \right) \left( \int \frac{\partial^2 \nu(f^*(x), g^*(x))}{\partial f^2(x)} (p_\lambda - f)^2 + \int \frac{\partial^2 \nu(f^*(x), g^*(x))}{\partial g^2(x)} (q_\lambda - g)^2 + \right.$$
$$\left. 2 \int \frac{\partial^2 \nu(f^*(x), g^*(x))}{\partial g(x) \partial g(x)} (p_\lambda - f)(q_\lambda - g) \right)$$
$$\geq \sigma_{\min} \left( \|p_\lambda - f\|^2 + \|q_\lambda - g\|^2 \right) \geq \sigma_{\min} \left( K_1^2 + K_2^2 \right)$$

For the second step note that $(f^*, g^*)$ lies in line segment between $(p_\lambda, q_\lambda)$ and $(f, g)$ and is therefore both upper and lower bounded. Therefore, the Hessian evaluated at $(f^*, g^*)$ is strictly positive definite with some minimum eigenvalue $\sigma_{min}$. For the third step we have used that $(p_\lambda - f, q_\lambda - g) = (K_1 \sum_{j=1}^{\ell_1} \gamma_j u_j, K_2 \sum_{j=1}^{\ell_2} \delta_j v_j)$ and that the $u_j$'s are orthonormal and $\|u_j\|_2 = 1$. This establishes the $2\beta$ separation between the null and the alternative as required by Theorem 20 with $\beta = \sigma_{\min}(K_1^2 + K_2^2)/2$. Precisely,

$$T(p_\lambda, q_\lambda) \geq T(f, g) + \mathcal{O}(\ell_1^{-2s/d} + \ell_2^{-2s/d})$$

Now we need to bound the Hellinger separation, between $F^n \times G^m$ and $\overline{P \times Q}$. First note that by our construction,

$$\overline{P \times Q} = \frac{1}{|\Lambda|} \sum_{\lambda \in \Lambda} P_\lambda^n \times Q_\lambda^m = \left( \frac{1}{|\Gamma|} \sum_{\gamma \in \Gamma} P_\gamma^n \right) \times \left( \frac{1}{|\Delta|} \sum_{\delta \in \Delta} Q_\delta^m \right) = \overline{P^n} \times \overline{Q^m}$$

By the tensorization property of the Hellinger affinity we have,

$$H^2(F^n \times G^m, \overline{P \times Q}) = 2 \left( 1 - \left( 1 - \frac{H^2(F^n, \overline{P^n})}{2} \right) \left( 1 - \frac{H^2(G^m, \overline{Q^m})}{2} \right) \right)$$
$$\leq H^2(F^n, \overline{P^n}) + H^2(G^m, \overline{Q^m})$$

We now apply Proposition 21 to bound each Hellinger divergence. If we denote $\rho_j(\cdot) = K_1 u_j(\cdot)/f(\cdot)$ then we see that the $\rho_j$'s satisfy the conditions of the proposition and further $p_\gamma = f(1 + \sum_j \gamma_j \rho_j)$ allowing us to use the bound. Accordingly $\alpha_j = \int \rho_j^2 f \leq C K_1^2 / \ell_1$ for some $C$. Hence,

$$H^2(F^n, \overline{P^n}) \leq \frac{n^2}{3} \sum_{j=1}^{m} \alpha_j^2 \leq \frac{C n^2 K_1^4}{\ell_1} \in \mathcal{O}(n^2 \ell_1^{-\frac{4s+d}{d}}).$$

A similar argument yields $H^2(G^m, \overline{Q^m}) \in \mathcal{O}(m^2 \ell_2^{-\frac{4s+d}{d}})$. If we pick $\ell_1 = n^{\frac{2d}{4s+d}}$ and $\ell_2 = m^{\frac{2d}{4s+d}}$ and hence $K_1 = n^{-\frac{2s}{2s+d}}$ and $K_2 = m^{-\frac{2s}{2s+d}}$, then we have that the Hellinger separation is bounded by a constant.

$$H^2(F^n \times G^m, \overline{P \times Q}) \leq H^2(F^n, \overline{P^n}) + H^2(G^m, \overline{Q^m}) \in \mathcal{O}(1)$$

Further, the error is larger than $\beta \asymp K_1^s + K_2^2 \asymp n^{\frac{-4s}{2s+d}} + m^{\frac{-4s}{2s+d}}$.

The first part of the lower bound for $\tau = 8s/(4s + d)$ is concluded by Markov's inequality,

$$\frac{\mathbb{E}[(\widehat{T} - T(f, g))^2]}{(n^{-\tau/2} + m^{-\tau/2})^2} \leq \mathbb{P}\left( |\widehat{T} - T(f, g)| > (n^{-\tau/2} + m^{-\tau/2}) \right) > c$$

where we note that $(n^{-\tau/2} + m^{-\tau/2})^2 \asymp n^{-\tau} + m^{-\tau}$. The $n^{-1} + m^{-1}$ lower bound is straightforward as as we cannot do better than the the parametric rate [2]. See [12] for an proof that uses a contradiction argument in the setting $n = m$. □

# F An Illustrative Example - The Conditional Tsallis Divergence

In this section we present a step by step guide on applying our framework to estimating any desired functional. We choose the Conditional Tsallis divergence because pedagogically it is a good example in Table 1 to illustrate the technique. By following a similar procedure, one may derive an estimator for any desired functional. The estimators are derived in Section F.1 and in Section F.2 we discuss conditions for the theoretical guarantees and asymptotic normality.

The Conditional Tsallis-$\alpha$ divergence ($\alpha \neq 0, 1$) between $X$ and $Y$ conditioned on $Z$ can be written in terms of joint densities $p_{XZ}, p_{YZ}$.

$$C_\alpha^T(p_{X|Z}\|p_{Y|Z}; p_Z) = C_\alpha^T(p_{XZ}, p_{YZ}) = \int p_Z(z) \frac{1}{\alpha - 1} \left( \int p_{X|Z}^\alpha(u, z) p_{Y|Z}^{1-\alpha}(u, z) \mathrm{d}u - 1 \right) \mathrm{d}z$$

$$= \frac{1}{1 - \alpha} + \frac{1}{\alpha - 1} \int p_{XZ}^\alpha(u, z) p_{YZ}^\beta(u, z) \mathrm{d}u \mathrm{d}z$$

where we have taken $\beta = 1 - \alpha$. We have samples $V_i = (X_i, Z_{1i}) \sim p_{XZ}, i = 1, \ldots, n$ and $W_j = (Y_j, Z_{1j}) \sim p_{YZ}, j = 1, \ldots, m$ We will assume $p_{XZ}, p_{YZ} \in \Sigma(s, L, B', B)$. For brevity, we will write $p = (p_{XZ}, p_{YZ})$ and $\hat{p} = (\widehat{p}_{XZ}, \widehat{p}_{YZ})$.

## F.1 The Estimators

We first compute the influence functions of $C_\alpha^T$ and the use it to derive the DS/LOO estimators.

**Proposition 23** (Influence Functions of $C_\alpha^T$). *The influence functions of $C_\alpha^T$ w.r.t $p_{XZ}, p_{YZ}$ are*

$$\psi_{XZ}(X, Z_1; p_{XZ}, p_{YZ}) = \frac{\alpha}{\alpha - 1} \left( p_{XZ}^{\alpha-1}(X, Z_1) p_{YZ}^\beta(X, Z_1) - \int p_{XZ}^\alpha p_{YZ}^\beta \right) \quad (29)$$

$$\psi_{YZ}(Y, Z_2; p_{XZ}, p_{YZ}) = - \left( p_{XZ}^\alpha(Y, Z_2) p_{YZ}^{\beta-1}(Y, Z_2) - \int p_{XZ}^\alpha p_{YZ}^\beta \right)$$

*Proof.* Recall that we can derive the influence functions via $\psi_{XZ}(X, Z_1; p) = C_{\alpha\ XZ}^{T'}(\delta_{X, Z_1} - p_{XZ}; p)$, $\psi_{YZ}(Y, Z_2; p) = C_{\alpha\ YZ}^{T'}(\delta_{X, Z_2} - p_{YZ}; p)$ where $C_{\alpha\ XZ}^{T'}, C_{\alpha\ YZ}^{T'}$ are the Gâteaux derivatives of $C_\alpha^T$ w.r.t $p_{XZ}, p_{YZ}$ respectively. Hence,

$$\psi_{XZ}(X, Z_1) = \frac{1}{\alpha - 1} \frac{\partial}{\partial t} \int ((1-t)p_{XZ} + t\delta_{XZ_1})^\alpha p_{YZ}^\beta \Big|_{t=0}$$

$$= \frac{\alpha}{\alpha - 1} \int p_{XZ}^{\alpha-1} p_{YZ}^\beta (\delta_{XZ_1} - p_{XZ})$$

from which the result follows. Deriving $\psi_{YZ}$ is similar. Alternatively, we can directly show that $\psi_{XZ}, \psi_{YZ}$ in Equation (29) satisfy Definition 2. $\square$

**DS estimator**: Use $V_1^{n/2}, W_1^{m/2}$ to construct density estimates $\widehat{p}_{XZ}^{(1)}, \widehat{p}_{YZ}^{(1)}$ for $p_{XZ}, p_{YZ}$. Then, use $V_{n/2+1}^{2n}, W_{m/2+1}^m$ to add the sample means of the influence functions given in Theorem 23. This results in our preliminary estimator,

$$\widehat{C}_\alpha^{T(1)} = \frac{1}{1 - \alpha} + \frac{\alpha}{\alpha - 1} \frac{2}{n} \sum_{i=n/2+1}^n \left( \frac{\widehat{p}_{XZ}^{(1)}(X_i, Z_{1i})}{\widehat{p}_{YZ}^{(1)}(X_i, Z_{1i})} \right)^{\alpha-1} - \frac{2}{m} \sum_{j=m/2+1}^m \left( \frac{\widehat{p}_{XZ}^{(1)}(Y_j, Z_{2j})}{\widehat{p}_{YZ}^{(1)}(Y_j, Z_{2j})} \right)^\alpha$$

(30)

The final estimate is $\widehat{C}_{\alpha,\mathrm{DS}}^T = (\widehat{C}_\alpha^{T(1)} + \widehat{C}_\alpha^{T(2)})/2$ where $\widehat{C}_\alpha^{T(2)}$ is obtained by swapping the two samples.

**LOO Estimator:** Denote the density estimates of $p_{XZ}, p_{YZ}$ without the $i^{\text{th}}$ sample by $\widehat{p}_{XZ,-i}$ and $\widehat{p}_{YZ,-i}$. Then the LOO estimator is,

$$\widehat{C}_{\alpha,\mathrm{LOO}}^T = \frac{1}{1 - \alpha} + \frac{\alpha}{\alpha - 1} \frac{1}{n} \sum_{i=1}^n \left( \frac{\widehat{p}_{XZ,-i}(X_i, Z_{1i})}{\widehat{p}_{YZ}(X_i, Z_{1i})} \right)^{\alpha-1} - \left( \frac{\widehat{p}_{XZ}(Y_i, Z_{2i})}{\widehat{p}_{YZ,-i}(Y_i, Z_{2i})} \right)^\alpha \quad (31)$$

### F.2 Analysis and Asymptotic Confidence Intervals

We begin with a functional Taylor expansion of $C_\alpha^T(f,g)$ around $(f_0, g_0)$. Since $\alpha, \beta \neq 0, 1$, we can bound the second order terms by $O\left(\|f - f_0\|^2 + \|g - g_0\|^2\right)$.

$$C_\alpha^T(f,g) = C_\alpha^T(f_0, g_0) + \frac{\alpha}{\alpha - 1} \int f_0^{\alpha-1} g_0^\beta - \int f_0^\alpha g_0^{\beta-1} + O\left(\|f - f_0\|^2 + \|g - g_0\|^2\right) \quad (32)$$

Precisely, the second order remainder is,

$$\frac{\alpha^2}{\alpha - 1} \int f_*^{\alpha-2} g_*^\beta (f - f_0)^2 - \beta \int f_*^\alpha g_*^{\beta-2} (g - g_0)^2 + \frac{\alpha\beta}{\alpha - 1} \int f_*^{\alpha-1} g_*^\beta (f - f_0)(g - g_0)$$

where $(f_*, g_*)$ is in the line segment between $(f, g)$ and $(f_0, g_0)$. If $f, g, f_0, g_0$ are bounded above and below so are $f_*, g_*$ and $f_*^a g_*^b$ where $a, b$ are coefficients depending on $\alpha$. The first two terms are respectively $O\left(\|f - f_0\|^2\right)$, $O\left(\|g - g_0\|^2\right)$. The cross term can be bounded via, $\left|\int (f - f_0)(g - g_0)\right| \leq \int \max\{|f - f_0|^2, |g - g_0|^2\} \in O(\|f - f_0\|^2 + \|g - g_0\|^2)$.

As mentioned earlier, the boundedness of the densities give us the required rates given in Theorems 7 for both estimators.

For the DS estimator, to show asymptotic normality, we need to verify the conditions in Theorem 19. We state it formally below, but prove it at the end of this section.

**Corollary 24.** *Let $p_{XY}, p_{XZ} \in \Sigma(s, L, B, B')$. Then $\widehat{C}_{\alpha,\mathrm{DS}}^T$ is asymptotically normal when $p_{XZ} \neq p_{YZ}$ and $s > d/2$.*

Finally, to construct a confidence interval we need a consistent estimate of the asymptotic variance : $\frac{1}{\zeta}\mathbb{V}_{XZ}\left[\psi_{XZ}(V; p)\right] + \frac{1}{1-\zeta}\mathbb{V}_{YZ}\left[\psi_{YZ}(W; p)\right]$ where,

$$\mathbb{V}_{XZ}\left[\psi_{XZ}(X, Z_1; p_{XZ}, p_{YZ})\right] = \left(\frac{\alpha}{\alpha - 1}\right)^2 \left(\int p_{XZ}^{2\alpha-1} p_{YZ}^{2\beta} - \left(\int p_{XZ}^\alpha p_{YZ}^\beta\right)^2\right)$$

$$\mathbb{V}_{YZ}\left[\psi_{YZ}(Y, Z_2; p_{XZ}, p_{YZ})\right] = \left(\int p_{XZ}^{2\alpha} p_{YZ}^{2\beta-1} - \left(\int p_{XZ}^\alpha p_{YZ}^\beta\right)^2\right)$$

From our analysis above, we know that any functional of the form $S(a, b) = \int p_{XZ}^a p_{YZ}^b, a + b = 1, a, b \neq 0, 1$ can be estimated via a LOO estimate

$$\widehat{S}(a, b) = \frac{1}{n}\sum_{i=1}^n a\frac{\widehat{p}_{YZ,-i}^b(V_i)}{\widehat{p}_{XZ,-i}^b(V_i)} + b\frac{\widehat{p}_{XZ,-i}^a(W_i)}{\widehat{p}_{YZ,-i}^a(W_i)}$$

where $\widehat{p}_{XZ,-i}, \widehat{p}_{YZ,-i}$ are the density estimates from $V_{-i}, W_{-i}$ respectively. $n/N$ is a consistent estimator for $\zeta$. This gives the following estimator for the asymptotic variance,

$$\frac{N}{n}\frac{\alpha^2}{(\alpha-1)^2}\widehat{S}(2\alpha - 1, 2\beta) + \frac{N}{m}\widehat{S}(2\alpha, 2\beta - 1) - \frac{N(m\alpha^2 + n(\alpha - 1)^2)}{nm(\alpha - 1)^2}\widehat{S}^2(\alpha, \beta).$$

The consistency of this estimator follows from the consistency of $\widehat{S}(a, b)$ for $S(a, b)$, Slutzky's theorem and the continuous mapping theorem.

*Proof of Corollary 24.* We now prove that the DS estimator satisfies the necessary conditions for asymptotic normality. We begin by showing that $C_\alpha^T$'s influence functions satisfy the regularity condition 4. We will show this for $\psi_{YZ}$. The proof for $\psi_{XZ}$ is similar. Consider two pairs of densities $(f, g)$ $(f', g')$ on the $(XZ, YZ)$ spaces.

$$\int \left(\psi_{XZ}(u; f, g) - \psi_{XZ}(u; f', g')\right)^2 f$$

$$= \frac{\alpha^2}{(1 - \alpha)^2} \int \left(f^{\alpha-1}g^\beta - \int f^\alpha g^\beta - \left[f'^{\alpha-1}g'^\beta - \int f'^\alpha g'^\beta\right]\right)^2 f$$

$$\leq 2\frac{\alpha^2}{(1-\alpha)^2}\left[\int\left(f^{\alpha-1}g^\beta - f'^{\alpha-1}g'^\beta\right)^2 f + \left(\int f^\alpha g^\beta - \int f'^\alpha g'^\beta\right)^2\right]$$

$$\leq 2\frac{\alpha^2}{(1-\alpha)^2}\left[\int\left(f^{\alpha-1}g^\beta - f'^{\alpha-1}g'^\beta\right)^2 f + \int\left(f^\alpha g^\beta - f'^\alpha g'^\beta\right)^2\right]$$

$$\leq 4\frac{\alpha^2}{(1-\alpha)^2}\Big[\|g^\beta\|_\infty^2\int(f^{\alpha-1}-f'^{\alpha-1})^2 + \|f'^{\alpha-1}\|_\infty^2\int(g^\beta-g'^\beta)^2 +$$

$$\|g^\beta\|_\infty^2\int(f^\alpha-f'^\alpha)^2 + \|f'^\alpha\|_\infty^2\int(g^\beta-g'^\beta)^2\Big]$$

$$\in O\left(\|f-f'\|^2\right) + O\left(\|g-g'\|^2\right)$$

where, in the second and fourth steps we have used Jensen's inequality. The last step follows from the boundedness of all our densities and estimates and by lemma 11.

The bounded variance condition of the influence functions also follows from the boundedness of the densities.

$$\mathbb{V}_{p_{XZ}}\psi_{XZ}(V; p_{XZ}, p_{YZ}) \leq \frac{\alpha^2}{(\alpha-1)^2}\mathbb{E}_{p_{XZ}}\left[p_{XZ}{}^{2\alpha-2}(X,Z_1)p_{YZ}{}^{2\beta}(X,Z_1)\right]$$

$$= \frac{\alpha^2}{(\alpha-1)^2}\int p_{XZ}{}^{2\alpha-1}p_{YZ}{}^{2\beta} < \infty$$

We can bound $\mathbb{V}_{p_{YZ}}\psi_{YZ}$ similarly. For the fourth condition, note that when $p_{XZ} = p_{YZ}$,

$$\psi_{XZ}(X,Z_1; p_{XZ}, p_{XZ}) = \frac{\alpha}{\alpha-1}\left(p_{XZ}{}^{\alpha+\beta-1}(X,Z_1) - \int p_{XZ}\right) = 0,$$

and similarly $\psi_{YZ} = \mathbf{0}$. Otherwise, $\psi_{XZ}$ depends explicitly on $X, Z$ and is nonzero. Therefore we have asymptotic normality away from $p_{XZ} = p_{YZ}$. $\qquad\square$

## G  Addendum to Experiments

### G.1  Details on Simulations

In our simulations, for the first figure comparing the Shannon Entropy in Fig 1 we generated data from the following one dimensional density,

$$f_1(t) = 0.5 + 0.5t^9$$

For this, with probability $1/2$ we sample from the uniform distribution $U(0,1)$ on $(0,1)$ and otherwise sample 10 points from $U(0,1)$ and pick the maximum. For the third figure in Fig 1 comparing the KL divergence, we generate data from the one dimensional density

$$f_2(t) = 0.5 + \frac{0.5t^{19}(1-t)^{19}}{B(20,20)}$$

where $B(\cdot,\cdot)$ is the Beta function. For this, with probability $1/2$ we sample from $U(0,1)$ and otherwise sample from a Beta$(20,20)$ distribution. The second and fourth figures of Fig 1 we sampled from a 2 dimensional density where the first dimension was $f_1$ and the second was $U(0,1)$. The fifth and sixth were from a 2 dimensional density where the first dimension was $f_2$ and the second was $U(0,1)$. In all figures of Fig. 2, the first distribution was a 4-dimensional density where all dimensions are $f_2$. The latter was $U(0,1)^4$.

**Methods compared to:**  In addition to the plug-in, DS and LOO estimators we perform comparisons with several other estimators. For the Shannon Entropy we compare our method to the $k$-NN estimator of Goria et al. [8], the method of Stowell and Plumbley [33] which uses $K-D$ partitioning, the method of Noughabi and Noughabi [23] based on Vasicek's spacing method and that of Learned-Miller and John [15] based on Voronoi tessellation. For the KL divergence we compare against the $k$-NN method of Pérez-Cruz [26] and that of Ramırez et al. [29] based on the power spectral density representation using Szego's theorem. For Rényi-$\alpha$ , Tsallis-$\alpha$ and Hellinger divergences we compared against the $k$-NN method of Póczos et al. [28].

$$(a) \qquad\qquad\qquad\qquad\qquad\qquad\qquad (b)$$

Figure 3: (a) Some sample images from the three categories apples, cows and cups. (b) The affinity matrix used in clustering.

### G.2 Image Clustering Task

Here we demonstrate a simple image clustering task using a nonparametric divergence estimator. For this we use images from the ETH-80 dataset. The objective here is not to champion our approach for image clustering against all methods for image clustering out there. Rather, we just wish to demonstrate that our estimators can be easily and intuitively applied to many Machine Learning problems.

We use the three categories Apples, Cows and Cups and randomly select $50$ images from each category. Some sample images are shown in Fig 3(a). We convert the images to grey scale and extract the SIFT features from each image. The SIFT features are $128$-dimensional but we project it to $4$ dimensions via PCA. This is necessary because nonparametric methods work best in low dimensions. Now we can treat each image as a collection of features, and hence a sample from a $4$ dimensional distribution. We estimate the Hellinger divergence between these "distributions". Then we construct an affinity matrix $A$ where the similarity metric between the $i^{\text{th}}$ and $j^{\text{th}}$ image is given by $A_{ij} = \exp(-\widehat{H}^2(X_i, X_j))$. Here $X_i$ and $X_j$ denotes the projected SIFT samples from images $i$ and $j$ and $\widehat{H}(X_i, X_j)$ is the estimated Hellinger divergence between the distributions. Finally, we run a spectral clustering algorithm on the matrix $A$.

Figure 3(b) depicts the affinity matrix $A$ when the images were ordered according to their class label. The affinity matrix exhibits block-diagonal structure which indicates that our Hellinger divergence estimator can in fact identify patterns in the images. Our approach achieved a clustering accuracy of $92.47\%$. When we used the $k$-NN based estimator of [28] we achieved an accuracy of $90.04\%$. When we instead applied Spectral clustering naively, with $A_{ij} = \exp(-L_2(P_i, P_j)^2)$ where $L_2(P_i, P_j)$ is the squared $L_2$ distance between the pixel intensities we achieved an accuracy of $70.18\%$. We also tried $A_{ij} = \exp(-\alpha \widehat{H}^2(X_i, X_j))$ as the affinity for different choices of $\alpha$ and found that our estimator still performed best. We also experimented with the Rényi-$\alpha$ and Tsallis-$\alpha$ divergences and obtained similar results.

On the same note, one can imagine that these divergence estimators can also be used for a classification task. For instance we can treat $\exp(-\widehat{H}^2(X_i, X_j))$ as a similarity metric between the images and use it in a classifier such as an SVM.

# H  Estimators for Some Information Theoretic Quantities

In this section, we present estimators for several common Information Theoretic quantities. The definitions and estimators are in Table 1. The table presents the LOO estimators.

For several functionals (e.g. conditional and unconditional Rényi-$\alpha$ divergence, conditional Tsallis-$\alpha$ mutual information and more) the estimators are not listed only because the expressions are too long to fit into the table. Our software implements a total of 17 functionals which include all the estimators in the table.

| Functional | LOO Estimator |
|---|---|
| Tsallis-$\alpha$ Entropy $\frac{1}{\alpha-1}\left(1-\int p^\alpha\right)$ | $\frac{1}{1-\alpha} + \frac{1}{n}\sum_i \int \widehat{p}_{-i}^\alpha - \frac{\alpha}{\alpha-1}\widehat{p}_{-i}^{\alpha-1}(X_i)$ |
| Rényi-$\alpha$ Entropy $\frac{-1}{\alpha-1}\log\int p^\alpha$ | $\frac{\alpha}{\alpha-1} + \frac{1}{n}\sum_i \frac{-1}{\alpha-1}\log\int \widehat{p}_{-i}^\alpha - \widehat{p}_{-i}^{\alpha-1}(X_i)+$ |
| Shannon Entropy $-\int p\log p$ | $-\frac{1}{n}\sum_i \log\widehat{p}_{-i}(X_i)$ |
| $L_2^2$ Divergence $\int (p_X-p_Y)^2$ | $\frac{2}{n}\sum_i \widehat{p}_{X,-i}(X_i) - \widehat{p}_Y(X_i) - \int(\widehat{p}_{X,-i}-\widehat{p}_Y)^2 + \frac{2}{m}\sum_j \widehat{p}_X(Y_j) - \widehat{p}_{Y,-j}(Y_j)$ |
| Hellinger Divergence $2-2\int p_X^{1/2}p_Y^{1/2}$ | $2 - \frac{1}{n}\sum_i \widehat{p}_{X,-i}^{-1/2}(X_i)\widehat{p}_Y^{1/2}(X_i) - \frac{1}{m}\sum_j \widehat{p}_X^{1/2}(Y_j)\widehat{p}_{Y,-j}^{-1/2}(Y_j)$ |
| Chi-Squared Divergence $\int \frac{(p_X-p_Y)^2}{p_X}$ | $-1 + \frac{1}{n}\sum_i \frac{\widehat{p}_Y^2(X_i)}{\widehat{p}_{X,-i}^2(X_i)} + 2\frac{1}{m}\sum_j \frac{\widehat{p}_{Y,-j}(Y_j)}{\widehat{p}_X(Y_j)}$ |
| $f$-Divergence $\int \phi(\frac{p_X}{p_Y})p_Y$ | $\frac{1}{n}\sum_i \phi'\left(\frac{\widehat{p}_{X,-i}(X_i)}{\widehat{p}_Y(X_i)}\right) + \frac{1}{m}\sum_j \left(\phi\left(\frac{\widehat{p}_{Y,-j}(Y_j)}{\widehat{p}_X(Y_j)}\right) - \frac{\widehat{p}_X(Y_j)}{\widehat{p}_{Y,-j}(Y_j)}\phi\left(\frac{\widehat{p}_X(Y_j)}{\widehat{p}_{Y,-j}(Y_j)}\right)\right)$ |
| Tsallis-$\alpha$ Divergence $\frac{1}{\alpha-1}\left(\int p_X^\alpha p_Y^{1-\alpha}-1\right)$ | $\frac{1}{1-\alpha} + \frac{\alpha}{\alpha-1}\frac{1}{n}\sum_i \left(\frac{\widehat{p}_{X,-i}(X_i)}{\widehat{p}_Y(X_i)}\right)^{\alpha-1} - \frac{1}{m}\sum_j \left(\frac{\widehat{p}_X(Y_j)}{\widehat{p}_{Y,-j}(Y_j)}\right)^\alpha$ |
| KL divergence $\int p_X\log\frac{p_X}{p_Y}$ | $1 + \frac{1}{n}\sum_i \log\frac{\widehat{p}_{X,-i}(X_i)}{\widehat{p}_Y(X_i)} - \frac{1}{m}\sum_j \frac{\widehat{p}_X(Y_j)}{\widehat{p}_{Y,-j}(Y_j)}$ |
| Conditional-Tsallis-$\alpha$ divergence $\int p_Z\frac{1}{\alpha-1}\left(\int p_{X|Z}^\alpha p_{Y|Z}^{1-\alpha}-1\right)$ | $\frac{1}{1-\alpha} + \frac{\alpha}{\alpha-1}\frac{1}{n}\sum_i \left(\frac{\widehat{p}_{XZ,-i}(V_i)}{\widehat{p}_{YZ}(V_i)}\right)^{\alpha-1} - \frac{1}{m}\sum_j \left(\frac{\widehat{p}_{XZ}(W_j)}{\widehat{p}_{YZ,-j}(W_j)}\right)^\alpha$ |
| Conditional-KL divergence $\int p_Z\int p_{X|Z}\log\frac{p_{X|Z}}{p_{Y|Z}}$ | $1 + \frac{1}{n}\sum_i \log\frac{\widehat{p}_{XZ,-i}(V_i)}{\widehat{p}_{YZ}(V_i)} - \frac{1}{m}\sum_j \frac{\widehat{p}_{XZ}(W_j)}{\widehat{p}_{YZ,-j}(W_j)}$ |
| Shannon Mutual Information $\int p_{XY}\log\frac{p_{XY}}{p_X p_Y}$ | $\frac{1}{n}\sum_i \log\widehat{p}_{XY,-i}(X_i,Y_i) - \log\widehat{p}_{X,-i}(X_i) - \log\widehat{p}_{Y,-i}(Y_i)$ |
| Conditional Tsallis-$\alpha$ MI $\int p_Z\frac{1}{\alpha-1}\left(\int p_{X,Y|Z}^\alpha p_{X|Z}^{1-\alpha}p_{Y|Z}^{1-\alpha}-1\right)$ | $\frac{1}{1-\alpha} + \frac{1}{\alpha-1}\frac{1}{n}\sum_i \alpha\left(\frac{\widehat{p}_{XYZ,-i}(X_i,Y_i,Z_i)\widehat{p}_Z(Z_i)}{\widehat{p}_{XZ,-i}(X_i,Z_i)\widehat{p}_{YZ,-i}(Y_i,Z_i)}\right)^{\alpha-1}$ $-(1-\alpha)\frac{1}{\alpha-1}\frac{1}{n}\sum_i \widehat{p}_Z^{\alpha-2}(Z_i)\int \widehat{p}_{XYZ,-i}^\alpha(\cdot,\cdot,Z_i)\widehat{p}_{XZ,-i}^{1-\alpha}(\cdot,Z_i)$ $+\frac{1}{\alpha-1}\frac{1}{n}\sum_i (1-\alpha)\widehat{p}_{XZ,-i}^{-\alpha}(X_i,Z_i)\widehat{p}_Z^{1-\alpha}(Z_i)\int \widehat{p}_{XYZ,-i}^\alpha(X_i,\cdot,Z_i)\widehat{p}_{YZ,-i}^{1-\alpha}(\cdot,Z_i)$ $+\frac{1}{\alpha-1}\frac{1}{n}\sum_i (1-\alpha)\widehat{p}_{YZ,-i}^{-\alpha}(Y_i,Z_i)\widehat{p}_Z^{1-\alpha}(Z_i)\int \widehat{p}_{XYZ,-i}^\alpha(\cdot,Y_i,Z_i)\widehat{p}_{XZ,-i}^{1-\alpha}(\cdot,\cdot)$ |

Table 1: Definitions of functionals and the corresponding estimators. Here $p_{X|Z}, p_{XZ}$ etc. are conditional and joint distributions. For the conditional divergences we take $V_i = (X_i, Z_{1i})$, $W_j = (Y_j, Z_{2j})$ to be the samples from $p_{XZ}, p_{YZ}$ respectively. For the mutual informations we have samples $(X_i, Y_i) \sim p_{XY}$ and for the conditional versions we have $(X_i, Y_i, Z_i) \sim p_{XYZ}$.