[Reviews · NeurIPS 2015]

Submitted by Assigned_Reviewer_1

In this paper, estimators for statistical functionals (of multiple distributions) are proposed, based on influence functions. A leave one out technique is proposed to estimate functionals of a single distribution with the same convergence rate as the data-split estimator. This is in turn used to estimate entropy, mutual info, etc. The results are complemented with experiments. I'm not familiar with Von Mises estimation, but to the best of my knowledge, the paper is quite well-written, interesting and original.

Minor comment: The authors may want to get their dictionaries consistent (either use an all UK, or an all US dictionary throughout).
Summary: In this paper, estimators for statistical functionals are proposed, based on influence functions. The results are complemented with experiments. The paper is quite well-written, interesting and original.

Submitted by Assigned_Reviewer_2

Summary of paper: This paper presents a leave-one-out (LOO) estimator for a functional of a single distribution and second LOO estimator for a functional of two distributions. The authors present analysis for these LOO estimators as well as for data split (DS) estimators. Lastly, the paper presents experiments on synthetic datasets and an benchmark dataset.

The authors have many goals for this paper from introducing new ideas to the machine learning community to proofs and experiments. It feels as though there could be two papers here: one for functionals of a single distribution and a second for functionals of two distributions. By splitting this into two papers, the authors would have more space in the main paper to add additional exposition detailing the purpose of each idea introduced, instead of relying so heavily on the supplemental material.

This paper is a challenging read. The authors have several moments of clarification that are a welcome break from the deluge of technical functional analytic details. For example, the last sentence of the first paragraph does an excellent job summarizing the majority of the paper. Further examples of helpful clarification are the discussions following the list of contributions (starting at line 75) and Theorem 8 (starting at line 308).

In contrast, at the top of the second page, the authors state that they will "introduce the calculus of influence functions to the machine learning community." While the authors do provide a technical definition for influence functions, they do not provide approachable exposition on influence functions, explaining what they are, what they can be used for, or why one may use them. Despite reading the paper several times, I had to seek out additional resources to gain a reasonable understanding of influence functions. There are several missed opportunities for short exposition about influence functions, such as at line 85 or following Definition 2 at line 111.

The writing style and organization of the paper conceal the machine learning advancements and how these advances differ from the previous work. For example, by placing the "Comparison with Other Approaches" at the near end of the paper, a reader may be mislead into thinking that the work presented in sections 2 through 4 are either the work of the authors or standard in this sub-field of influence functions. After reading section 5, I was aware of the differences between this work and previous work and thus, I was better prepared to re-read sections 2 through 4 than I was after reading those sections the first time. I recommend both moving section 5 to earlier in the paper (either as part of section 1 or immediately following section 1) and to add a bit of exposition at the beginning of section 3 "Estimating Functionals" that makes it clear where the authors contributions begin.

There are many impressive facets of this work. The results on the ETH-80 dataset are impressive in context of the previous work presented in this paper. I wish there had been additional exposition about this experiment in the main paper. I also like that they present an open question (at line 267) that is a result of their work. This makes it clear that their work has pushed the boundary for the field and has opened further research avenues.

Summary: This paper attempts to accomplish a lot in just 8 pages and, as a result, fluctuates between being very approachable with excellent exposition adding to readers' understanding of the work and extremely technical with little exposition or context for the presented technical details. To make this paper more readable and to clarify the contributions of this paper in the context of previous work, I think that this paper needs some reorganization and proofreading, along with additional clarifying exposition.

Submitted by Assigned_Reviewer_3

The authors propose new nonparametric estimators for information theoretic quantities, based on techniques from the semiparametric statistics literature.

The estimators are based on a "leave-one-out" (LOO) technique, as opposed to previous estimators, which are based on data-splitting.

The LOO estimators have similar theoretical properties to the data-splitting estimators, but seem to have better empirical performance. The authors provide convergence rates for the LOO estimators and derive new asymptotic normality results for the data-splitting estimators.

The performance of the proposed estimators is illustrated in some empirical examples

I think that topics/techniques discussed in this paper cover an area where theoretical statistics may be able to make a significant contribution to machine learning.

Thus, I think that papers like this should be encouraged at conferences like NIPS.

However, I also think that this paper has some limitations.

It would be nice if the authors were able to obtain an asymptotic normality result for the LOO estimator (not just the data-splitting estimator). Also in this direction, it would be good if the authors could give a convincing concrete example of the practical usefulness of asymptotic normality results in the present context.

Additionally, the authors claim that their methods do "not require any tuning of hyperparameters"; however, it appears that tuning is required to implement the KDE estimators that form the basis of their methods.

(I understand that it may be more difficult to tune other methods, but it seems like tuning parameters could still be an issue for the methods proposed here.)
Summary: The authors use techniques from the semiparametric statistics literature to derive new nonparametric estimators of information theoretic quantities and corresponding convergence results.

I think this is a nice application of classical techniques from statistics in an area of interest for machine learning.

Author Feedback
Author rebuttal: We thank all reviewers for their comments.

"No hyper-parameters to tune":
As some of you have pointed out what we meant here was that we have a good automatic
procedure for selecting the kde bandwidth -namely cross validation on the density
estimate. We wanted to emphasise that there are no principled way to choose the
hyper parameters in most competing methods that give the correct rate of converge.
We shall paraphrase this properly in the paper.

In all experiments, the bandwidth was chosen via cross validation (line 377).

@ Reviewer 1:
- We agree with most of your comments on our claims and the references. We shall make
necessary changes and cite the papers.

@ Reviewer 2:
- Yes, the paper is somewhat dense and the introduction to IFs is brief. We will try
to improve on the explanations a bit more.
- We see that your other comments are minor details and typos. Thanks for pointing
them out. We shall fix them.

@ Reviewer 3:
- Asymptotic normality is a standard method in Statistics for constructing
confidence intervals and hypothesis testing.

@ Reviewer 7:
- Even though the asymptotic rates are the same in theory, practical performance may
vary if the constants are large especially with few samples.